# You Need Better Attention Priors

Elon Litman [1]   Gabe Guo [1]

 **github.com/elonlit/goat**

## Abstract

We generalize the attention mechanism by viewing it through the lens of Entropic Optimal Transport, revealing that standard attention corresponds to a transport problem regularized by an implicit uniform prior. We introduce Generalized Optimal transport Attention with Trainable priors (GOAT 🐐), a new attention mechanism that replaces this naive assumption with a learnable, continuous prior. This prior maintains full compatibility with optimized kernels such as FlashAttention. GOAT also provides an EOT-based explanation of attention sinks and materializes a solution for them, avoiding the representational trade-offs of standard attention. Finally, by absorbing spatial information into the core attention computation, GOAT learns an extrapolatable prior that combines the flexibility of learned positional embeddings with the length generalization of fixed encodings.

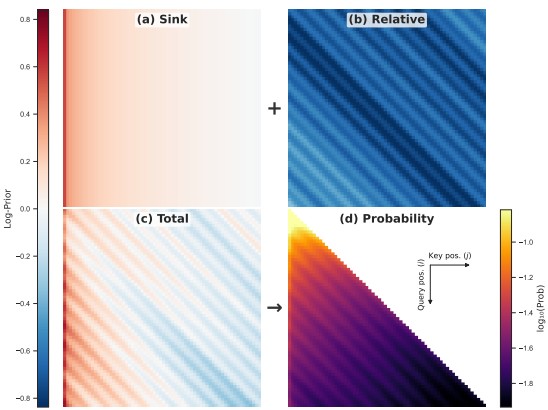

*Figure 1.* **Learned GOAT log-prior decomposition on a toy copy-mixture task.** *Task:* train a small causal LM on synthetic sequences where each token is a mixture of copying the first token (global), copying the previous token (local), or random noise. **(a)** Key-only sink component $K_{\text{sink}}(i, j) = u(j)$. **(b)** Translation-equivariant relative component $K_{\text{rel}}(i, j) = \kappa(i - j)$. **(c)** Row-centered total log-prior $K$ (row shifts do not affect softmax). **(d)** Induced causal prior probabilities after masking to $j \leq i$ and row-normalizing with softmax (axes shown: query position $i$, key position $j$).

## 1. Introduction

Self-attention is the core computational primitive of the Transformer architecture (Vaswani et al., 2017; Bahdanau et al., 2015; Luong et al., 2015), driving state-of-the-art performance in language modeling (Brown et al., 2020), vision (Dosovitskiy et al., 2021), generative modeling with diffusion processes (Ho et al., 2020), and sequential decision making (Chen et al., 2021). While empirically successful, scaled dot-product attention is still largely motivated by heuristics: the dot product serves as an intuitive similarity measure, and the softmax is treated as a smooth `argmax` surrogate. This has motivated a large body of work that refines the original formulation along three main axes: computational efficiency, expressivity, and behavioral stability. On

the efficiency side, linear-time approximations (Katharopoulos et al., 2020; Choromanski et al., 2021), sparse attention patterns (Kitaev et al., 2020; Beltagy et al., 2020; Zaheer et al., 2020), and I/O-aware kernels such as FlashAttention (Dao, 2023; Shah et al., 2024) have become standard practice. To improve expressivity, many methods inject inductive biases, for example, locality and relative positioning, directly into the attention scores via relative position encodings and attention biases (Shaw et al., 2018; Dai et al., 2019; Su et al., 2021; Press et al., 2022). More recently, studies of model stability have identified emergent behaviors such as attention sinks, where models allocate disproportionate mass to a small set of tokens in long-context regimes (Xiao et al., 2024; Chen et al., 2024). Our work revisits the mathematical foundations of attention to provide a unified perspective that addresses all three axes, yielding a generalized mechanism that is efficient, expressive, and offers direct control over such emergent behaviors.

[1]Department of Computer Science, Stanford University. Correspondence to: Elon Litman <elonlit@stanford.edu>.

*Proceedings of the 43$^{rd}$ International Conference on Machine Learning*, Seoul, South Korea. PMLR 306, 2026. Copyright 2026 by the author(s).

Recent work reframes self-attention through the lens of Entropic Optimal Transport (EOT) (Litman, 2025; Cuturi, 2013; Sinkhorn & Knopp, 1967). In this view, the standard attention weights arise as the transport plan that distributes a Dirac delta impulse throughout the key space while being maximally entropic. We suggest an equivalent perspective: the tendency toward maximal entropy can be regarded as the tendency to remain as close as possible, in the Kullback–Leibler (KL) sense, to a uniform prior over positions. Interpreting attention as a KL-regularized transport problem suggests a broader family of mechanisms: replacing the implicit uniform prior with structured priors induces a richer, more controllable class of attention rules. In this light, we view standard PEs as heuristic approximations of a platonic EOT-derived prior, arguing that their lack of a unified derivation leads to structural entanglement and instability. We pursue this idea and make four core contributions.

First, we formalize this generalization and show that incorporating an arbitrary prior distribution $\pi$ into the EOT objective yields a simple closed-form solution: for each query, the optimal attention distribution is a softmax over content scores shifted by the log-prior,

$$p^\star = \mathrm{softmax}(s/\tau + \log \pi), \qquad (1)$$

where $\tau$ is a temperature parameter and $s$ collects the unscaled dot-product scores. Structurally, the log-prior additively contours the transport costs, providing a mechanism for encoding inductive biases directly into the objective rather than into the content representations.

Second, we introduce GOAT, a highly efficient mechanism that realizes this structure within a single, unmodified scaled dot-product attention (SDPA) call. By factorizing query and key vectors into content and positional subspaces, GOAT absorbs the log-prior into the dot-product logits while preserving head dimension, maintaining full compatibility with optimized kernels such as FlashAttention and its successors (Dao, 2023) without materializing dense bias matrices. The module is initialized to either a uniform prior (recovering standard attention) or a maximum-entropy recency prior (approximating ALiBi), so that complex spatial biases are only learned when they improve upon this baseline.

Third, we demonstrate that GOAT improves compute efficiency and outperforms baselines on long-context retrieval, computer vision, and computational biology tasks. Our method significantly reduces peak memory usage while maintaining robust generalization to sequence lengths far beyond those seen during training.

Finally, we leverage the KL-prior EOT formulation to give a formal account of *attention sinks* (Xiao et al., 2024; Gu et al., 2024), interpreting them as the natural outcome of the EOT objective on low-signal queries under a peaked prior. The GOAT construction turns this phenomenon into

a controllable modeling primitive: by factorizing keys into dedicated subspaces, we inject a key-only bias into the log-prior, yielding a disentangled way to control sink behavior without corrupting content representations.

## 2. Preliminaries: Attention as Entropic Optimal Transport

The core of our approach is the variational interpretation of the attention mechanism, which reinterprets the attention weights as the solution to a one-sided EOT problem (Litman, 2025). Consider a single query $i$ as a unit impulse of mass (a Dirac delta $\delta_i$) that must be distributed across the sequence of keys $\{j\}_{j=1}^L$. The cost of transporting mass from query $i$ to key $j$ is defined by their negative affinity, $c_{ij} = -s_{ij}$. Unlike classical doubly constrained OT, attention enforces only the query-side simplex constraint; no marginal constraint is imposed over keys. The attention mechanism seeks a transport plan $p \in \Delta^{L-1}$ that minimizes the expected transport cost while maintaining high entropy.

**Definition 2.1** (The EOT Objective). The attention weights $p^\star$ are the unique minimizer of the transport cost regularized by Shannon entropy:

$$p^\star = \arg \min_{p \in \Delta^{L-1}} \left\{ \underbrace{\langle p, -s \rangle}_{\text{Transport Cost}} - \underbrace{\tau H(p)}_{\text{Regularization}} \right\}, \qquad (2)$$

where $s$ is the vector of unscaled dot-product scores, $H(p) \triangleq -\sum_j p_j \log p_j$ is the Shannon entropy (Shannon, 1948), and $\tau > 0$ is the temperature.

**Proposition 2.2.** *The solution to* (2) *recovers the standard softmax attention mechanism. We provide the full derivation in Appendix B.*

This derivation reveals that standard attention is the unique distribution that is maximally non-committal (i.e., highest entropy) subject to matching the expected score. The Shannon entropy regularizer, $H(p)$, can be seen as penalizing the deviation from a uniform distribution. A natural generalization is to replace this with a regularizer that penalizes deviation from an arbitrary *prior distribution* $\pi \in \Delta^{L-1}$. The Kullback-Leibler (KL) divergence provides the measure for this (Kullback & Leibler, 1951).

## 3. Generalizing the Attention Prior

The derivation in Section 2 reveals a critical limitation of standard attention. The Shannon entropy regularizer, $H(p)$, can be equivalently viewed as the negative Kullback-Leibler divergence between $p$ and a fixed uniform distribution $\mathcal{U}$:

$$-H(p) = \mathrm{KL}(p \,\|\, \mathcal{U}) - \log L. \qquad (3)$$

Consequently, standard attention assumes an uninformative, flat prior over the sequence. It penalizes any deviation from

uniformity that is not justified by the content scores.

We generalize this framework by replacing the naive uniform assumption with an arbitrary prior distribution $\boldsymbol{\pi} \in \Delta^{L-1}$. This prior encodes structural expectations—such as locality, periodicity, or specific sinks—directly into the transport objective.

**Proposition 3.1** (Attention with Priors). *Let $\boldsymbol{\pi}$ be a fixed prior distribution over keys. We generalize the regularization term from Shannon entropy to the Kullback-Leibler divergence* $\mathrm{KL}(\boldsymbol{p} \,\|\, \boldsymbol{\pi})$. *The optimal transport plan is:*

$$\boldsymbol{p}^{\star} = \arg \min_{\boldsymbol{p} \in \Delta^{L-1}} \left\{ -\langle \boldsymbol{p}, \boldsymbol{s} \rangle + \tau \mathrm{KL}(\boldsymbol{p} \,\|\, \boldsymbol{\pi}) \right\}. \quad (4)$$

*The unique solution to this problem is a softmax over scores shifted by the log-prior:*

$$p_j^{\star} = \mathrm{softmax} \left( \frac{s_j}{\tau} + \log \pi_j \right). \quad (5)$$

*This result formally identifies the missing term in the attention mechanism: standard PEs are merely heuristic approximations of this EOT-derived prior* $\log \pi$.

*Proof.* We expand the KL divergence term in the objective function $J(\boldsymbol{p})$:

$$\mathrm{KL}(\boldsymbol{p} \,\|\, \boldsymbol{\pi}) = \sum_j p_j \log \frac{p_j}{\pi_j} \quad (6)$$

$$= \sum_j p_j \log p_j - \sum_j p_j \log \pi_j. \quad (7)$$

Substituting this back into Equation 4 and grouping terms linear in $p_j$:

$$\begin{aligned} J(\boldsymbol{p}) = & -\sum_j p_j s_j \\ & + \tau \sum_j p_j \log p_j - \tau \sum_j p_j \log \pi_j \\ = & \sum_j p_j \underbrace{(-s_j - \tau \log \pi_j)}_{\text{Effective Cost}} + \tau \underbrace{\sum_j p_j \log p_j}_{-H(\boldsymbol{p})}. \quad (8) \end{aligned}$$

This transforms the KL-regularized problem into a standard Shannon-regularized problem (Equation 2) with a modified cost vector. The effective score for token $j$ becomes $s_j + \tau \log \pi_j$. The optimal probability is proportional to the exponential of the effective score:

$$p_j^{\star} \propto \exp \left( \frac{s_j + \tau \log \pi_j}{\tau} \right) = \exp \left( \frac{s_j}{\tau} + \log \pi_j \right) \quad (9)$$

$$= \exp \left( \frac{s_j}{\tau} \right) \exp(\log \pi_j) = \pi_j \exp(s_j/\tau). \quad (10)$$

Normalizing this result according to Equation 39 yields the solution in Equation 5. □

This perspective reveals standard attention as the special case of a uniform prior. While the theory permits arbitrary distributions, we prove in Appendix G that imposing practical constraints such as SDPA-compatibility, translation equivariance, and stability uniquely restricts the admissible priors to a finite trigonometric family. The remainder of this paper is dedicated to developing an expressive and computationally efficient method to learn this prior.

## 4. Parameterizing the Prior With GOAT

We have established that the standard attention mechanism is the unique solution to an EOT problem regularized by a uniform prior. By generalizing the regularizer to the Kullback-Leibler divergence against an arbitrary prior $\boldsymbol{\pi}$, we derived the optimal attention distribution. Identifying the temperature $\tau$ with the standard scaling factor $\sqrt{d_c}$ yields:

$$p_{ij} = \mathrm{softmax}_j \left( \frac{\langle \boldsymbol{q}_{c,i}, \boldsymbol{k}_{c,j} \rangle}{\sqrt{d_c}} + \mathcal{K}_{ij} \right), \quad (11)$$

where $s_{ij} = \langle \boldsymbol{q}_{c,i}, \boldsymbol{k}_{c,j} \rangle / \sqrt{d_c}$ represents the content-based affinity and $\mathcal{K}_{ij}$ is an unnormalized log-prior (the softmax normalizes $\exp(\mathcal{K})$ into a proper prior).

The structural inductive biases of the attention mechanism are determined entirely by the parameterization of $\mathcal{K}_{ij}$. We introduce Generalized Optimal transport Attention with Trainable priors (GOAT). GOAT parameterizes $\mathcal{K}_{ij}$ as a continuous, differentiable function of token positions that satisfies three criteria: it must express shift-invariant (relative) relationships including directionality, it must support global defaults (attention sinks), and it must be computationally realizable within standard attention kernels (e.g., FlashAttention) without materializing $L \times L$ bias matrices.

**Spectral Decomposition of Relative Position** We first consider the shift-invariant component of the prior. This inductive bias ensures that attention patterns depend solely on relative distance, allowing learned structures to generalize to sequence positions unseen during training. We parameterize this log-prior using a truncated Fourier series (Rahimi & Recht, 2007). While motivated by the spectral representation of shift-invariant kernels (see Appendix D), our formulation operates on unnormalized logits. This allows the spectral weights $\alpha_r, \beta_r$ to take negative values, enabling the model to learn not just local periodicity (attraction) but also explicit suppression patterns (repulsion) at specific frequencies. To capture both symmetric and asymmetric relationships, we define:

$$\mathcal{K}_{ij}^{\mathrm{rel}} = \sum_{r=1}^{R} \left[ \alpha_r \cos(\omega_r(i-j)) + \beta_r \sin(\omega_r(i-j)) \right]. \quad (12)$$

where $\{\omega_r\}_{r=1}^{R}$ are fixed geometric frequencies. The coefficients $\alpha_r$ and $\beta_r$ are learnable spectral weights, where

$\alpha_r$ controls symmetric interactions, while $\beta_r$ controls anti-symmetric interactions. In all experiments, unless otherwise stated, we use base $B = 10{,}000$ and set

$$\omega_r = B^{-(r-1)/\max(R-1,1)}, \qquad r = 1, \ldots, R. \quad (13)$$

To incorporate this prior efficiently, we must linearize Equation (12) into an inner product of query and key vectors. We apply the angle difference identities:

$$\cos(\omega_r(i-j)) = \cos(\omega_r i)\cos(\omega_r j) \qquad (14)$$
$$+ \sin(\omega_r i)\sin(\omega_r j), \qquad (15)$$
$$\sin(\omega_r(i-j)) = \sin(\omega_r i)\cos(\omega_r j) \qquad (16)$$
$$- \cos(\omega_r i)\sin(\omega_r j). \qquad (17)$$

Substituting these into Equation 12 allows us to factorize the expression. We define a positional subspace of dimension $d_r = 2R$. For the $r$-th frequency, we define the positional key vector $\boldsymbol{k}_{\text{rel},j}^{(r)} \in \mathbb{R}^2$ simply as the Fourier feature of position $j$:

$$\boldsymbol{k}_{\text{rel},j}^{(r)} = \begin{bmatrix} \cos(\omega_r j) & \sin(\omega_r j) \end{bmatrix}^\mathsf{T}. \quad (18)$$

The corresponding query vector $\boldsymbol{q}_{\text{rel},i}^{(r)} \in \mathbb{R}^2$ is constructed by a spectral rotation parameterized by $\alpha_r$ and $\beta_r$:

$$\boldsymbol{q}_{\text{rel},i}^{(r)} = \begin{bmatrix} \alpha_r \cos(\omega_r i) + \beta_r \sin(\omega_r i) \\ \alpha_r \sin(\omega_r i) - \beta_r \cos(\omega_r i) \end{bmatrix}. \quad (19)$$

We provide the full derivation verifying that this dot product recovers the desired spectral term in Appendix C. By concatenating these components across all $R$ frequencies, we obtain vectors $\boldsymbol{q}_{\text{rel},i}$ and $\boldsymbol{k}_{\text{rel},j}$ whose inner product reconstructs $\mathcal{K}_{ij}^{\text{rel}}$.

**Extension to 2D positions.** For image patches, we apply the same construction axis-wise. If token $i$ has grid coordinate $(y_i, x_i)$, we split the $R$ frequencies into vertical and horizontal groups and define

$$\mathcal{K}_{ij}^{\text{2D}} = \sum_{r \in \mathcal{R}_y} \alpha_r \cos(\omega_r(y_i - y_j))\beta_r \sin(\omega_r(y_i - y_j))$$
$$+ \sum_{r \in \mathcal{R}_x} \alpha_r \cos(\omega_r(x_i - x_j))\beta_r \sin(\omega_r(x_i - x_j)). \quad (20)$$

This is still an exact query-key dot product: the positional feature is the concatenation of the 1D Fourier features for the two axes. For multimodal sequences, image tokens receive canonical 2D coordinates while text tokens keep their 1D causal positions, so different heads can learn modality-appropriate priors inside the same attention layer.

**Explicit sink parameterization.** A pervasive phenomenon in Transformer models is the emergence of *attention sinks*: the tendency to allocate substantial probability mass to specific tokens, often the initial token, regardless of the query, particularly when no other token is semantically relevant (Xiao et al., 2024). Standard attention typically induces sinks by learning high-norm *content* keys, entangling a structural default with semantic representation. We instead introduce a dedicated key-only logit bias $u(j)$ in the prior, as mandated by our EOT analysis (Section 5). We parameterize $u_h(j)$ for each head as a learned key-linear recency term plus a small absolute-position MLP and an optional first-token bump:

$$u_h(j) = \lambda_h j + g_h(\phi_{\text{abs}}(j)) + b_h \mathbf{1}\{j = 0\}. \quad (21)$$

Here $\phi_{\text{abs}}(j)$ concatenates $M$ sinusoidal features with $j/L_{\text{train}}$ and $\log(1 + j)/\log(1 + L_{\text{train}})$. The MLP $g_h$ is a one-hidden-layer SiLU network initialized near zero. Implementation uses one additional dimension in the SDPA dot product:

$$\langle \boldsymbol{q}_{\text{sink},i}, \boldsymbol{k}_{\text{sink},j} \rangle = \langle 1, u(j) \rangle = u(j) \qquad \forall i, \quad (22)$$

*i.e.*, a broadcast bias on key $j$. This yields an explicit query-independent default (e.g., $j = 0$) that dominates in low-signal regimes without corrupting content representations.

**Unified GOAT Parameterization** The full log-prior is the sum of the relative and absolute components: $\mathcal{K}_{ij} = \mathcal{K}_{ij}^{\text{rel}} + u(j)$. We realize this sum within a single attention operation by constructing composite vectors. Let $d_h$ be the total head dimension. We reserve a subspace of size $d_p = 2R + 2$ (padded for alignment) for the prior and use the remaining $d_c = d_h - d_p$ dimensions for content. This separation permits the query and key projection matrices to be constructed as block-diagonal, ensuring strict independence between semantic and structural subspaces.

Standard attention implementations (e.g., PyTorch, FlashAttention) compute the kernel $\text{softmax}(\boldsymbol{q}^\top \boldsymbol{k}/\sqrt{d_h})$. To inject our additive prior term $\mathcal{K}_{ij}$ without scaling it by $1/\sqrt{d_h}$, we must pre-scale the input vectors. We construct the composite query $\boldsymbol{q}_i'$ and key $\boldsymbol{k}_j'$ as follows:

$$\boldsymbol{q}_i' = \begin{bmatrix} \boldsymbol{q}_{c,i}\sqrt{\frac{d_h}{d_c}} & \boldsymbol{q}_{\text{rel},i}\sqrt{d_h} & \sqrt{d_h} \end{bmatrix}^\mathsf{T}, \quad (23)$$
$$\boldsymbol{k}_j' = \begin{bmatrix} \boldsymbol{k}_{c,j} & \boldsymbol{k}_{\text{rel},j} & u(j) \end{bmatrix}^\mathsf{T}. \quad (24)$$

When the standard dot-product attention kernel is applied to these vectors, the result is:

$$\frac{\langle \boldsymbol{q}_i', \boldsymbol{k}_j' \rangle}{\sqrt{d_h}} = \frac{1}{\sqrt{d_h}}\left( \sqrt{\frac{d_h}{d_c}}\langle \boldsymbol{q}_{c,i}, \boldsymbol{k}_{c,j} \rangle + \sqrt{d_h}\mathcal{K}_{ij} \right) \quad (25)$$
$$= \frac{\langle \boldsymbol{q}_{c,i}, \boldsymbol{k}_{c,j} \rangle}{\sqrt{d_c}} + \mathcal{K}_{ij} \quad (26)$$
$$= s_{ij} + \log \pi_{ij}. \quad (27)$$

Note that the content scores are scaled by $1/\sqrt{d_c}$ while the prior term $\mathcal{K}_{ij}$ enters unscaled (effective temperature 1). This is a deliberate design choice: it prevents the prior from being attenuated at high head dimensions and ensures stable structural biases.

**The Necessity of Disentanglement (Prior vs. PE).** Our EOT formulation establishes that the attention mechanism is incomplete without an explicit prior $\boldsymbol{\pi}$, which manifests as an additive term in the logits. While this is isomorphic to an additive positional encoding, the EOT derivation mandates a specific parameterization that avoids the pitfalls of heuristic encodings. Standard methods like RoPE inject position multiplicatively via rotation, yielding $z_{ij} = (R_i \boldsymbol{q})^\top (R_j \boldsymbol{k})$. This creates *structural entanglement*: the magnitude of the positional bias is coupled to the semantic norms $\|\boldsymbol{q}\|\|\boldsymbol{k}\|$. To enforce a structural default (e.g. an attention sink), such models must artificially inflate content vectors, corrupting the semantic representation. By strictly adding the log-prior, GOAT ensures *disentanglement*, allowing the model to optimally allocate mass to sinks in low-signal regimes without corrupting the content. Furthermore, this perspective frames existing PEs as limited heuristic approximations of the true prior. While biases like ALiBi correctly adopt additivity, they enforce a rigid, monotonic structure. GOAT replaces these heuristics with a learnable prior derived from the EOT objective. This yields a mechanism that is not only disentangled but fully expressive, capable of discovering complex structural dependencies, modeling stable defaults, while retaining the extrapolation robustness of fixed encodings.

This formulation allows GOAT to be implemented as a drop-in replacement for standard Multi-Head Attention (Algorithm 1), leveraging optimized I/O-aware kernels without modification or computational overhead. The value vectors $\boldsymbol{v}_j$ remain purely content-based and utilize the full head dimension, ensuring that spatial information influences only the routing weights and not the mixed representations.

## 5. Attention Sinks in the EOT View

Attention sinks are tokens that absorb probability mass when the query contains little semantic signal. While often viewed as a learned artifact necessary to satisfy the softmax constraint, our EOT formulation offers a first-principles explanation: sinks are the optimal solution to the KL-regularized objective in low-signal regimes.

Throughout this section, we fix a query $i$ and denote the total logit for key $j$ as $z_{ij} = s_{ij} + \mathcal{K}_{ij}$, where $s_{ij}$ is the content score and $\mathcal{K}_{ij}$ is the unnormalized log-prior.

**The Inevitability of a Default** The EOT objective provides a justification for why a default attention pattern must exist. When the content-based evidence $\boldsymbol{s}_i$ is weak or am-

---

**Algorithm 1** GOAT Forward Pass

---

1: **Input:** Content vectors $\boldsymbol{q}_c, \boldsymbol{k}_c, \boldsymbol{v}$; Indices $i, j$
2: **Parameters:** Spectral weights $\alpha, \beta$; Sink MLP $\phi$
3: **Define:** $d_h$ (head dim), $d_c$ (content dim)
4: // 1. Generate Prior Components
5: $\boldsymbol{q}_{\text{rel}} \leftarrow \text{SpectralRotate}(i, \alpha, \beta)$
6: $\boldsymbol{k}_{\text{rel}} \leftarrow \text{FourierFeat}(j)$
7: $u(j) \leftarrow \phi(\text{SinusoidalEnc}(j))$
8: // 2. Compose Vectors with Scaling Trick
9: $\boldsymbol{q}' \leftarrow \begin{bmatrix} \boldsymbol{q}_c \cdot \sqrt{d_h/d_c} \\ \boldsymbol{q}_{\text{rel}} \cdot \sqrt{d_h} \\ \sqrt{d_h} \\ 0 \end{bmatrix}, \quad \boldsymbol{k}' \leftarrow \begin{bmatrix} \boldsymbol{k}_c \\ \boldsymbol{k}_{\text{rel}} \\ u(j) \\ 0 \end{bmatrix}$
10: // 3. Compute Attention via Optimized Kernel
11: **return** FLASHATTENTION$(\boldsymbol{q}', \boldsymbol{k}', \boldsymbol{v})$

---

biguous (high entropy), the KL divergence term dominates the objective, forcing the posterior distribution to converge to the prior distribution.

**Theorem 5.1** (Collapse to Prior). *Fix a query $i$. Let $\boldsymbol{\pi}_i$ be the normalized prior distribution derived from $\mathcal{K}_i$. Let $\omega_i \triangleq \max_k s_{ik} - \min_k s_{ik}$ be the dynamic range of the content scores. The posterior probability $p_{ij}$ satisfies:*

$$\pi_{ij} \exp(-\omega_i) \leq p_{ij} \leq \pi_{ij} \exp(\omega_i). \tag{28}$$

*Consequently, in the limit of low content signal where $\omega_i \to 0$, the posterior converges pointwise to the prior: $\lim_{\omega_i \to 0} \boldsymbol{p}_i = \boldsymbol{\pi}_i$. (See Section E.1 for full derivation).*

This theorem implies that every attention head must revert to its prior when semantic signal is lost. The distinction between robust models and unstable models lies in the shape of this prior.

**Formalizing Sinks via Margins** To ensure stability, the prior $\boldsymbol{\pi}$ must be sharply peaked, rather than uniform. We formalize this using the concept of a logit margin.

**Definition 5.2** (Sink by Margin). For a query $i$, a key $j^\star$ is an *attention sink* with margin $m_i(j^\star)$ if:

$$m_i(j^\star) \triangleq \min_{k \neq j^\star} (z_{ij^\star} - z_{ik}) > 0. \tag{29}$$

A positive margin guarantees a lower bound on probability mass, decoupling the sink's stability from the sequence length. However, the margin decomposes into two sources:

$$z_{ij^\star} - z_{ik} = \underbrace{(s_{ij^\star} - s_{ik})}_{\text{Content}} + \underbrace{(\mathcal{K}_{ij^\star} - \mathcal{K}_{ik})}_{\text{Prior}}. \tag{30}$$

**Standard Attention (Content Sink):** Since the implicit prior is uniform, $\mathcal{K}_{ij} = -\log L$. Creating a sink requires

$(s_{ij^\star} - s_{ik}) > 0$. The model must learn a generic key vector $\boldsymbol{k}_{c,j^\star}$ with large norm to force a high dot product across all queries. This entangles the structural role of the sink with the semantic representation of token $j^\star$.

**GOAT (Prior Sink):** Our method allows the creation of a sink via the second term. By learning a large key-specific bias $u(j^\star)$, we ensure $u(j^\star) - u(k) > 0$. This establishes a robust default without constraining the content vectors $\boldsymbol{k}_c$. The unscaled parameterization (Equation 27) effectively guarantees this preference by providing a gradient signal $\sqrt{d_c}$ times larger for defaults than the content channel.

**Stability and Total Context Sensitivity** We now quantify the benefit of a prior-driven sink by analyzing the stability of the output vector, $\boldsymbol{o}_i = \sum_j p_{ij} \boldsymbol{v}_j$, which is the weighted sum of value vectors. To do this, we measure the output's response to a small perturbation, $\Delta\boldsymbol{v}_k$, applied to the value vector of each context token $k \in \mathcal{C} = \{k \mid k \neq j^\star\}$. Ideally, the output should be invariant to such noise.

**Definition 5.3** (Total Context Sensitivity). For a given query $i$ and sink token $j^*$, we define the context $\mathcal{C}$ as the set of all other tokens in the sequence. The sensitivity of the output to the value vectors from this context is the total probability mass allocated to it:

$$\Psi(\mathcal{C}) \triangleq \sum_{k \in \mathcal{C}} p_{ik} = 1 - p_{ij^\star}. \tag{31}$$

Equivalently, if we perturb the context value vectors such that $\|\Delta\boldsymbol{v}_k\|_2 \leq \varepsilon$ for all $k \in \mathcal{C}$ while keeping the query and key vectors fixed, the change in the output is bounded:

$$\|\Delta\boldsymbol{o}_i\|_2 \leq \sum_{k \in \mathcal{C}} p_{ik} \|\Delta\boldsymbol{v}_k\|_2 \leq \varepsilon \sum_{k \in \mathcal{C}} p_{ik} = \varepsilon\, \Psi(\mathcal{C}). \tag{32}$$

The following theorem demonstrates that standard attention is asymptotically unstable, while GOAT suppresses sensitivity exponentially.

**Theorem 5.4** (Context Sensitivity Bounds). *Consider a sequence of length $L$ smaller than the prior's period. Let $\mathcal{C}$ be the set of context tokens excluding the sink $j^\star$. We analyze the sensitivity in the low-signal limit ($\omega_i \to 0$). (See Section E.2 for full derivation).*

1. *__Uniform Prior (Standard):__ With a uniform prior, sensitivity converges to 1 as sequence length increases:*

$$\lim_{L \to \infty} \lim_{\omega_i \to 0} \Psi_{uni}(\mathcal{C}) = 1. \tag{33}$$

2. *__Peaked Prior (GOAT):__ If the prior establishes a sink margin $\delta = \min_{k \in \mathcal{C}}(\mathcal{K}_{ij^\star} - \mathcal{K}_{ik}) > 0$, the sensitivity can be bounded arbitrarily low for any finite $L$:*

$$\lim_{\omega_i \to 0} \Psi_{sink}(\mathcal{C}) \leq \frac{L - 1}{\exp(\delta) + L - 1}. \tag{34}$$

Theorem 5.4 reveals a critical scaling advantage: while the sensitivity of standard attention converges to 1 as $L \to \infty$ (dominated by context noise), GOAT bounds it away from 1, suppressing noise exponentially with the prior margin $\delta$. Consequently, maintaining stability requires only logarithmic margin growth ($\delta \sim \ln L$). This is trivially achievable via the unconstrained bias $u(j)$, allowing the model to decouple sink behavior from semantic representation.

## 6. Experiments

This section is designed to empirically validate three core claims: (1) GOAT achieves the best of both worlds, combining the expressivity of learnable priors with the length extrapolation robustness of static encodings, surpassing both rigid heuristics (like RoPE) and non-generalizing learned embeddings, (2) GOAT validates the EOT prediction that attention sinks are optimal transport defaults, allowing them to be modeled explicitly and decoupled from semantic content, and (3) GOAT is a general mechanism applicable to diverse data modalities and structures, not just 1D sequences.

**Ablating the prior** To validate GOAT's ability to learn disentangled priors, we use a synthetic copy-mixture task where the optimal strategy is to attend to either the first token $j = 0$ or the previous token $j = i - 1$. Figure 1 demonstrates a successful recovery of the dominant ground-truth structure: the sink component (a) captures the global preference for $j = 0$, while the relative component (b) learns a periodic diagonal bias peaking at $j = i - 1$. The deep negative spectral troughs (blue bands) actively suppress attention to intervening positions, isolating the target diagonal candidate from the background noise, while the content-based mechanism resolves the remaining periodicity.

**Language Modeling and Extrapolation** We train 125M parameter models on the C4 dataset (Raffel et al., 2020). As shown in Figure 2, GOAT lowers in-distribution perplexity by *1.55 points* over ALiBi while maintaining robust extrapolation capabilities where RoPE degrades. We provide mode-count, frequency-base, initialization, learnable-frequency, and convergence ablations in Appendix I.

**Scaling to Billion-Parameter Language Models** To test whether the extrapolation benefits persist beyond the 125M setting, we train matched 1.23B-parameter Llama-style decoder models on FineWeb with $L_{\text{train}} = 2048$. The runs use identical optimizer, schedule, batch size, precision, and training window; the only architectural difference is the attention prior. As shown in Table 1, GOAT matches RoPE in-distribution while substantially improving extrapolation beyond the training context. RoPE deteriorates rapidly at $4\times$ and $8\times$ the training length, whereas GOAT degrades smoothly.

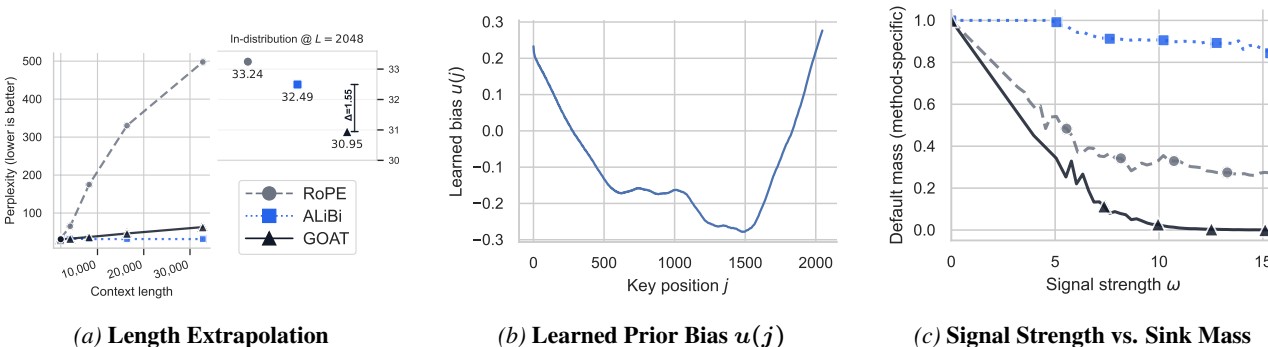

**(a) Length Extrapolation**      **(b) Learned Prior Bias $u(j)$**      **(c) Signal Strength vs. Sink Mass**

*Figure 2.* **GOAT combines the flexibility of learnable priors with robust length extrapolation.** Comparison of 125M parameter models trained on 4B tokens of C4 with context length $L_{\text{train}} = 2048$. **(a)** Length Extrapolation. Perplexity evaluated on extended sequences. RoPE degrades catastrophically past $L_{\text{train}}$. While ALiBi maintains flat extrapolation, it underfits the training window; the inset reveals GOAT improves in-distribution perplexity by **1.55** points over ALiBi. GOAT provides the best trade-off: superior fidelity within the window and robust generalization to 16× length. **(b)** Learned Prior Bias $u(j)$. The model spontaneously discovers a sharp spike at $j = 0$ (an explicit attention sink) and a rise at $j \approx 2000$ (local recency), validating that these structural needs can be decoupled from content. **(c)** Signal Strength vs. Sink Mass (Theorem 5.1). As signal $\omega$ increases, GOAT smoothly sheds sink mass from $\approx 1$ (prior-dominated) to $\approx 0$ (content-driven), while ALiBi remains over-defaulted (fixed distance bias adds an $\mathcal{O}(md)$ margin) and RoPE only partially disengages.

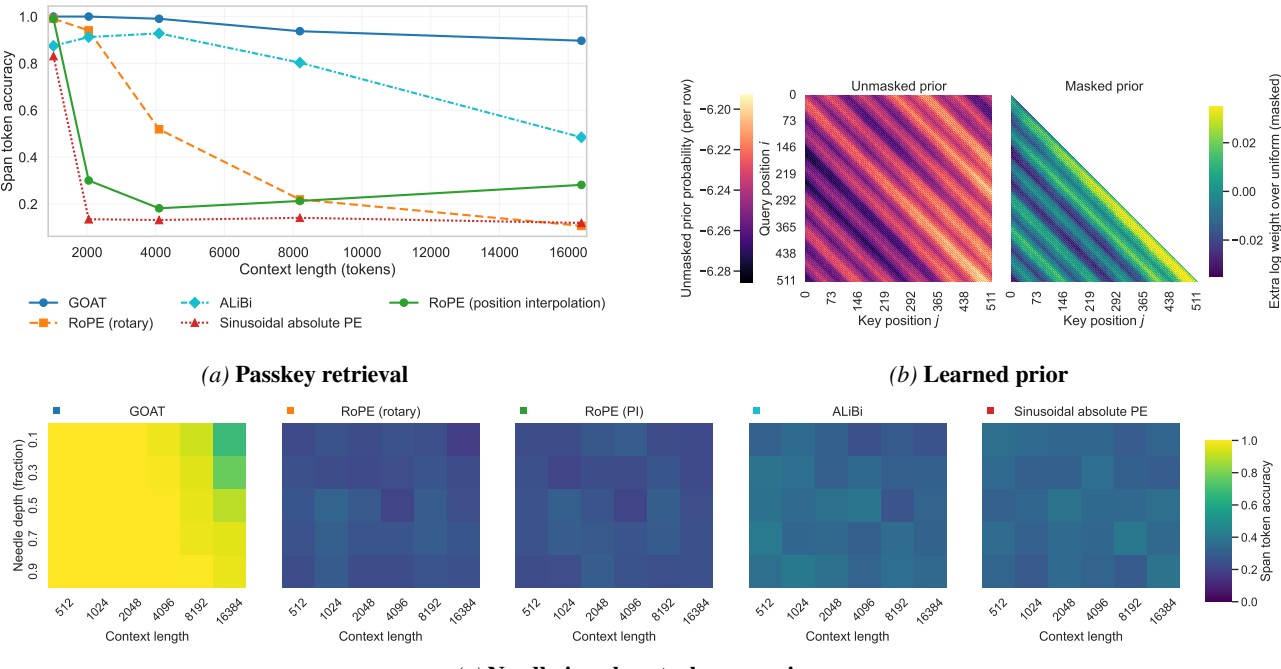

**(a) Passkey retrieval**                      **(b) Learned prior**

**(c) Needle-in-a-haystack comparison**

*Figure 3.* **Comparison of GOAT to other methods on long-context synthetic tasks. (a)** Span-token accuracy on the passkey retrieval task as a function of context length. Each model is trained as a GPT-style language model on sequences up to the training context length (e.g., $L_{\text{train}} = 1024$) and evaluated on an 8-digit passkey that appears once in the context and must be reproduced at the end of the sequence. The learned prior (GOAT) maintains substantially higher accuracy than rotary (Su et al., 2021), position-interpolated rotary (Chen et al., 2023), and sinusoidal absolute position encodings as the context length grows far beyond the training window. **(b)** Visualization of the learned log-prior for a GOAT attention head on a 512-token sequence. The left panel shows the unmasked prior, which allocates larger probability mass to later key positions; the right panel shows the same prior after applying the causal mask and row-renormalization, yielding a strong recency bias along the causal diagonal. This illustrates how the prior implements an explicit, interpretable inductive bias without modifying content scores. **(c)** Needle-in-a-haystack (NIAH) results: span-token accuracy heatmaps over context length and needle depth (fractional position of the needle) for the same attention variants. The learned prior maintains near-perfect retrieval across depths and lengths, while the rotary, interpolated rotary, and sinusoidal baselines degrade sharply as sequences become longer and the needle moves deeper into the context.

**Long-Context Retrieval** While perplexity measures overall capability, we further evaluate retrieval using long-context synthetic benchmarks (Liu et al., 2023). As shown

in Figure 3, GOAT maintains near-perfect accuracy on both Passkey Retrieval and Needle-in-a-Haystack (NIAH) tasks at context lengths far exceeding training, whereas other

*Table 1.* **Scaling to 1.23B Parameters on FineWeb.** GOAT matches RoPE at the training length and remains stable under length extrapolation, while RoPE degrades rapidly beyond the training window.

| Model | 2k | 4k | 8k | 16k |
|-------|-------|-------|-------|--------|
| RoPE | 20.82 | 28.53 | 89.45 | 417.43 |
| GOAT | **20.75** | **21.04** | **22.69** | **24.77** |

methods degrade catastrophically. ALiBi is especially brittle on NIAH because its fixed slopes impose large negative logits on distant needles in the steepest heads; GOAT can instead reduce the prior margin and return to content-driven retrieval when the needle is semantically decisive.

**Biological Sequence Modeling** We evaluate GOAT against a RoPE baseline on next-token language modeling of human-reference genome sequences, showing improvements and parity on validation NLL, training throughput, and qualitative nucleotide completions (Figure 4).

**Learning Priors on Image Data** To demonstrate the universality of the mechanism beyond 1D sequences, we apply GOAT to Vision Transformers on ImageNet-1k, finding that it spontaneously learns a 2D shift-invariant prior that enables robust zero-shot extrapolation to higher input resolutions (Figure 5).

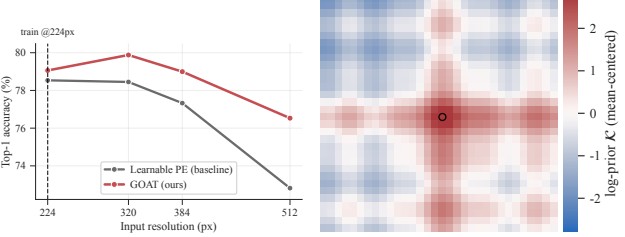

*(a)* **Resolution extrapolation**    *(b)* **Learned log-prior** $\mathcal{K}_{ij}$

*Figure 5.* GOAT **enables robust zero-shot resolution extrapolation on ImageNet-1k** (Deng et al., 2009). Models are trained at $224 \times 224$ and evaluated at higher resolutions without fine-tuning. **(a)** Top-1 accuracy vs. input resolution: the baseline ViT with absolute positional embeddings (Dosovitskiy et al., 2021) degrades as resolution increases, while GOAT maintains substantially higher accuracy. **(b)** Learned log-prior $\mathcal{K}_{ij}$ (shown relative to the center patch) exhibits a local, shift-invariant inductive bias despite uniform initialization.

## 7. Conclusion

This work identifies that the fragility of standard self-attention, manifesting as poor length generalization and emergent attention sinks, stems from the implicit assumption of a uniform prior within the mechanism's EOT formulation. We introduced Generalized Optimal transport

Attention with Trainable priors (GOAT), a mechanism that replaces this naive assumption with a learnable, expressive prior.

GOAT achieves three core advances. First, it resolves the trade-off between expressivity and stability, combining the robust length extrapolation of static heuristics like RoPE and ALiBi with the adaptability of learnable priors. Second, it provides a theoretical justification for attention sinks as optimal transport defaults that goes beyond empirical studies of sinks, allowing them to be modeled explicitly rather than entangled with semantic representations. Finally, GOAT realizes these benefits within standard I/O-aware kernels, offering a drop-in improvement for modern Transformers with minimal computational overhead.

## Impact Statement

This paper presents work whose goal is to advance the field of Machine Learning. There are many potential societal consequences of our work, none which we feel must be specifically highlighted here.

## Acknowledgments

We thank Stefano Ermon for his helpful feedback and discussions.

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

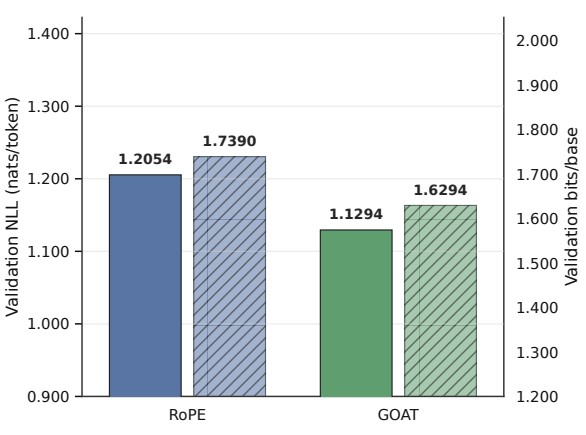

*(a)* **Validation NLL** (lower is better).

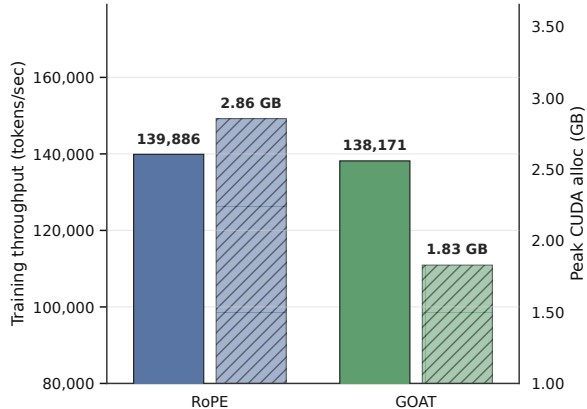

*(b)* **Computational Efficiency** (Throughput and Memory).

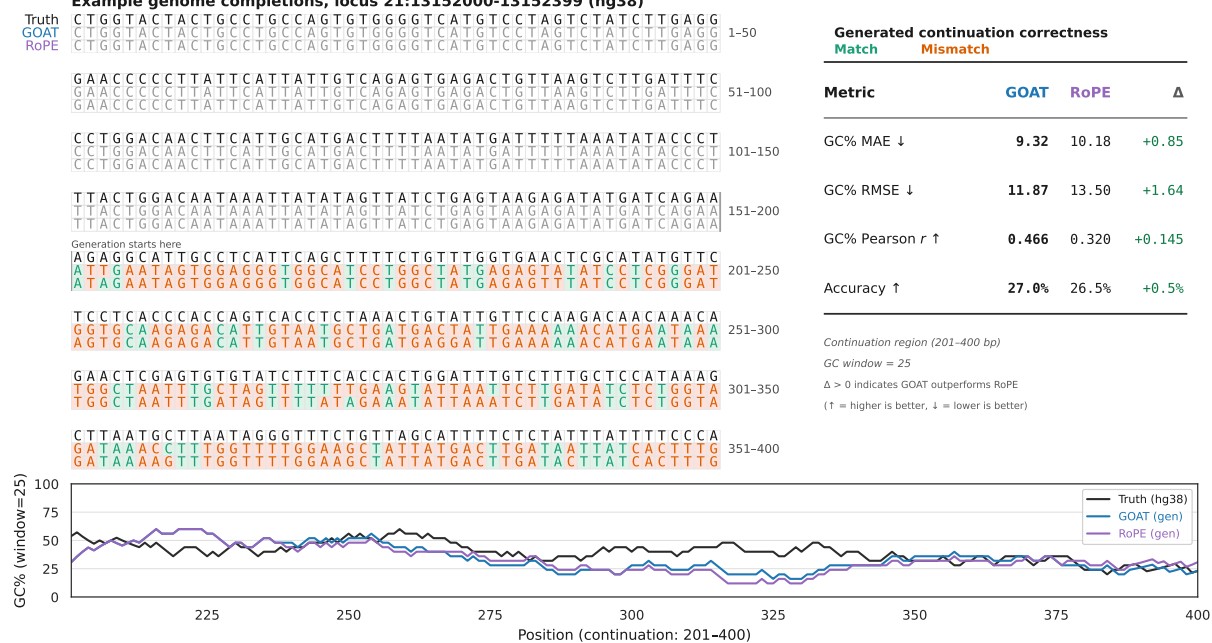

*(c)* **Example genome completions.** Prompt shown first, followed by the 200-nt continuation.

*Figure 4.* **GOAT outperforms RoPE on DNA modeling. (a)** Validation Negative Log-Likelihood (NLL) on the Human Reference Genome (125M parameter decoder-only LM). GOAT achieves lower Validation NLL (solid bars, left axis) and bits/base (hatched bars, right axis) than RoPE under identical training budgets. **(b)** Computational Efficiency. While training throughput (solid bars, left axis) remains comparable, GOAT significantly reduces Peak CUDA memory allocation (hatched bars, right axis), dropping from 2.86 GB to 1.83 GB (36% reduction). **(c)** Generative Quality. *Top:* Representative completion example and metrics table. The visualization shows a single sample, while reported metrics are averaged over $N = 3$ continuations. Generated nucleotides are colored by alignment to the ground truth (green = match, orange = mismatch). *Bottom:* Sliding-window GC% trajectory (window=25) for the generated continuation. GOAT (blue) tracks the ground-truth statistical profile (black) more accurately than RoPE (purple), evidenced by a higher Pearson correlation ($r = 0.466$ vs. 0.320).

Language Models. *arXiv preprint arXiv:2403.06764*, 2024.

Chen, S., Wong, S., Chen, L., and Tian, Y. Extending Context Window of Large Language Models via Positional Interpolation. *arXiv preprint arXiv:2306.15595*, 2023. doi: 10.48550/arXiv.2306.15595.

Choromanski, K., Likhosherstov, V., Dohan, D., Song, X.,

Gane, A., Sarlós, T., Hawkins, P., Davis, J., Mohiuddin, A., Kaiser, L., Belanger, D., Colwell, L., and Weller, A. Rethinking Attention with Performers. In *International Conference on Learning Representations*, 2021.

Choromanski, K. M., Li, S., Likhosherstov, V., Dubey, K. A., Luo, S., He, D., Yang, Y., Sarlos, T., Weingarten, T., and Weller, A. Learning a fourier transform for linear

relative positional encodings in transformers, 2024. URL https://arxiv.org/abs/2302.01925.

Cuturi, M. Sinkhorn Distances: Lightspeed Computation of Optimal Transport. In Burges, C. J. C., Bottou, L., Ghahramani, Z., and Weinberger, K. Q. (eds.), *Advances in Neural Information Processing Systems 26 (NIPS 2013)*, pp. 2292–2300, 2013.

Dai, Z., Yang, Z., Yang, Y., Carbonell, J., Le, Q. V., and Salakhutdinov, R. Transformer-XL: Attentive Language Models Beyond a Fixed-Length Context. In *Proceedings of the 57th Annual Meeting of the Association for Computational Linguistics*, 2019.

Dao, T. Flashattention-2: Faster attention with better parallelism and work partitioning. *arXiv preprint arXiv:2307.08691*, 2023.

Deng, J., Dong, W., Socher, R., Li, L.-J., Li, K., and Fei-Fei, L. ImageNet: A large-scale hierarchical image database. In *Proceedings of the IEEE Conference on Computer Vision and Pattern Recognition*, pp. 248–255, 2009. doi: 10.1109/CVPR.2009.5206848.

Dosovitskiy, A., Beyer, L., Kolesnikov, A., Weissenborn, D., Zhai, X., Unterthiner, T., Dehghani, M., Minderer, M., Heigold, G., Gelly, S., Uszkoreit, J., and Houlsby, N. An Image is Worth 16x16 Words: Transformers for Image Recognition at Scale. In *International Conference on Learning Representations*, 2021.

Gu, X., Pang, T., Du, C., Liu, Q., Zhang, F., Du, C., Wang, Y., and Lin, M. When Attention Sink Emerges in Language Models: An Empirical View, 2024.

Ho, J., Jain, A., and Abbeel, P. Denoising Diffusion Probabilistic Models. In *Advances in Neural Information Processing Systems*, volume 33, 2020.

Jaynes, E. T. Information Theory and Statistical Mechanics. *Physical Review*, 106(4):620–630, 1957. doi: 10.1103/PhysRev.106.620.

Katharopoulos, A., Vyas, A., Pappas, N., and Fleuret, F. Transformers are RNNs: Fast Autoregressive Transformers with Linear Attention. In *Proceedings of the 37th International Conference on Machine Learning*. PMLR, 2020.

Kitaev, N., Kaiser, Ł., and Levskaya, A. Reformer: The Efficient Transformer. In *International Conference on Learning Representations*, 2020. doi: 10.48550/arXiv.2001.04451.

Kullback, S. and Leibler, R. A. On Information and Sufficiency. *The Annals of Mathematical Statistics*, 22(1):79–86, 1951. doi: 10.1214/aoms/1177729694.

Litman, E. Scaled-Dot-Product Attention as One-Sided Entropic Optimal Transport. *arXiv preprint arXiv:2508.08369*, August 2025. doi: 10.48550/arXiv.2508.08369.

Liu, N. F., Lin, K., Hewitt, J., Paranjape, A., Bevilacqua, M., Petroni, F., and Liang, P. Lost in the Middle: How Language Models Use Long Contexts. *arXiv preprint arXiv:2307.03172*, 2023.

Luong, M.-T., Pham, H., and Manning, C. D. Effective Approaches to Attention-based Neural Machine Translation. In *Proceedings of the 2015 Conference on Empirical Methods in Natural Language Processing (EMNLP)*, pp. 1412–1421, 2015. doi: 10.18653/v1/D15-1166.

Press, O., Smith, N. A., and Lewis, M. Train Short, Test Long: Attention with Linear Biases Enables Input Length Extrapolation. In *International Conference on Learning Representations*, 2022.

Raffel, C., Shazeer, N., Roberts, A., Lee, K., Narang, S., Matena, M., Zhou, Y., Li, W., and Liu, P. J. Exploring the Limits of Transfer Learning with a Unified Text-to-Text Transformer. *Journal of Machine Learning Research*, 21 (140):1–67, 2020.

Rahimi, A. and Recht, B. Random features for large-scale kernel machines. In *Advances in Neural Information Processing Systems*, 2007.

Rasmussen, C. E. and Williams, C. K. I. *Gaussian Processes for Machine Learning*. MIT Press, 2006.

Shah, J., Bikshandi, G., Zhang, Y., Thakkar, V., Ramani, P., and Dao, T. FlashAttention-3: Fast and Accurate Attention with Asynchrony and Low-precision. *arXiv preprint arXiv:2407.08608*, 2024.

Shannon, C. E. A Mathematical Theory of Communication. *Bell System Technical Journal*, 27:379–423, 623–656, 1948.

Shaw, P., Uszkoreit, J., and Vaswani, A. Self-Attention with Relative Position Representations. In *Proceedings of the 2018 Conference of the North American Chapter of the Association for Computational Linguistics: Human Language Technologies*, pp. 464–468, 2018.

Sinkhorn, R. and Knopp, P. Concerning Nonnegative Matrices and Doubly Stochastic Matrices. *Pacific Journal of Mathematics*, 21(2):343–348, 1967.

Su, J., Lu, Y., Pan, S., Murtadha, A., Wen, B., and Liu, Y. RoFormer: Enhanced Transformer with Rotary Position Embedding, 2021.

Vaswani, A., Shazeer, N., Parmar, N., Uszkoreit, J., Jones, L., Gomez, A. N., Kaiser, Ł., and Polosukhin, I. Attention Is All You Need. In *Advances in Neural Information Processing Systems*, volume 30, pp. 5998–6008, 2017.

Xiao, G., Tian, Y., Chen, B., Han, S., and Lewis, M. Efficient streaming language models with attention sinks. In *International Conference on Learning Representations*, 2024. doi: 10.48550/arXiv.2309.17453. arXiv:2309.17453.

Zaheer, M., Guruganesh, G., Dubey, A., Ainslie, J., Alberti, C., Ontañón, S., Pham, P., Ravula, A., Wang, Q., Yang, L., and Ahmed, A. Big Bird: Transformers for Longer Sequences. In *Advances in Neural Information Processing Systems*, 2020. doi: 10.48550/arXiv.2007.14062.

# A. Related Work

**Relative position encodings and attention biases.**   The closest methods to GOAT are additive relative-bias schemes. Shaw et al. (2018) and T5 (Raffel et al., 2020) learn position-dependent logit shifts, but represent them with finite tables or buckets whose extrapolation behavior is not determined by the parameterization. RoPE (Su et al., 2021) gives strong length generalization, but it injects position by rotating semantic query and key vectors, coupling the strength of the positional effect to content norms. ALiBi (Press et al., 2022) preserves additivity and extrapolates by construction, but commits every head to a fixed monotone recency prior. The FLT method of Choromanski et al. (2024) also uses Fourier features for relative position, but targets linear attention. GOAT instead learns a continuous additive log-prior for standard softmax attention and realizes it exactly through the SDPA dot product.

**Attention sinks and long-context stability.**   Recent work on streaming and long-context transformers shows that attention sinks act as default destinations for probability mass (Xiao et al., 2024; Gu et al., 2024; Chen et al., 2024). These defaults are usually learned implicitly through content representations or enforced through fixed positional heuristics. Our formulation treats sinks as part of the prior itself: a key-only term can establish the default without requiring a semantic key vector to play a structural role.

**Optimal transport views of attention.**   Our derivation builds on entropy-regularized transport (Cuturi, 2013; Sinkhorn & Knopp, 1967) and the one-sided EOT view of scaled dot-product attention (Litman, 2025). In this view, standard attention is the special case induced by a uniform prior. GOAT keeps the same transport solution form, but replaces that implicit prior with a learned, structured one.

# B. Proof of Attention as One-Sided Entropic Optimal Transport

*Proof.* The objective is strictly convex. We introduce a Lagrange multiplier $\lambda$ for the mass constraint $\sum_j p_j = 1$. The Lagrangian is:

$$\mathcal{L}(\boldsymbol{p}, \lambda) = -\sum_{j=1}^{L} p_j s_j + \tau \sum_{j=1}^{L} p_j \log p_j$$
$$+ \lambda \left( \sum_{j=1}^{L} p_j - 1 \right). \tag{35}$$

The first-order stationarity condition with respect to $p_j$ is:

$$\frac{\partial \mathcal{L}}{\partial p_j} = -s_j + \tau(\log p_j + 1) + \lambda = 0. \tag{36}$$

Solving for $p_j$:

$$\log p_j = \frac{s_j - \lambda}{\tau} - 1 \implies \tag{37}$$

$$p_j = \exp\left(\frac{s_j}{\tau}\right) \exp\left(\frac{-\lambda}{\tau} - 1\right). \tag{38}$$

The second exponential term is a query-specific constant determined by the constraint $\sum p_j = 1$. Enforcing this constraint yields:

$$p_j^\star = \frac{\exp(s_j/\tau)}{\sum_k \exp(s_k/\tau)} \tag{39}$$

which is precisely the softmax function.                                                  □

# C. Factorization of the Spectral Prior

In Section 4, we introduced a parameterization of the relative log-prior $\mathcal{K}_{ij}^{\text{rel}}$ using a truncated Fourier series. We claimed that the inner product of specific query and key vectors recovers the term $\alpha_r \cos(\omega_r(i-j)) + \beta_r \sin(\omega_r(i-j))$. Here, we provide the explicit derivation.

Recall the definitions of the positional query component $q_{\text{rel},i}^{(r)}$ and key component $k_{\text{rel},j}^{(r)}$ for a specific frequency index $r$:

$$q_{\text{rel},i}^{(r)} = \begin{bmatrix} \alpha_r \cos(\omega_r i) + \beta_r \sin(\omega_r i) \\ \alpha_r \sin(\omega_r i) - \beta_r \cos(\omega_r i) \end{bmatrix}, \tag{40}$$

$$k_{\text{rel},j}^{(r)} = \begin{bmatrix} \cos(\omega_r j) \\ \sin(\omega_r j) \end{bmatrix}. \tag{41}$$

We compute the dot product $\langle q_{\text{rel},i}^{(r)}, k_{\text{rel},j}^{(r)} \rangle$ by expanding the element-wise multiplication:

$$\langle q_{\text{rel},i}^{(r)}, k_{\text{rel},j}^{(r)} \rangle = \Big( \alpha_r \cos(\omega_r i) + \beta_r \sin(\omega_r i) \Big) \cos(\omega_r j)$$
$$+ \Big( \alpha_r \sin(\omega_r i) - \beta_r \cos(\omega_r i) \Big) \sin(\omega_r j). \tag{42}$$

Next, we distribute the terms and regroup them by the coefficients $\alpha_r$ and $\beta_r$:

$$\langle q_{\text{rel},i}^{(r)}, k_{\text{rel},j}^{(r)} \rangle = \alpha_r \cos(\omega_r i) \cos(\omega_r j) + \beta_r \sin(\omega_r i) \cos(\omega_r j)$$
$$+ \alpha_r \sin(\omega_r i) \sin(\omega_r j) - \beta_r \cos(\omega_r i) \sin(\omega_r j)$$

$$= \alpha_r \Big[ \cos(\omega_r i) \cos(\omega_r j) + \sin(\omega_r i) \sin(\omega_r j) \Big]$$
$$+ \beta_r \Big[ \sin(\omega_r i) \cos(\omega_r j) - \cos(\omega_r i) \sin(\omega_r j) \Big]. \tag{43}$$

We apply the standard trigonometric angle difference identities:

$$\cos(A - B) = \cos A \cos B + \sin A \sin B, \tag{44}$$
$$\sin(A - B) = \sin A \cos B - \cos A \sin B. \tag{45}$$

Substituting these identities with $A = \omega_r i$ and $B = \omega_r j$, we obtain:

$$\langle q_{\text{rel},i}^{(r)}, k_{\text{rel},j}^{(r)} \rangle = \alpha_r \cos(\omega_r i - \omega_r j) + \beta_r \sin(\omega_r i - \omega_r j)$$
$$= \alpha_r \cos \big( \omega_r (i - j) \big) + \beta_r \sin \big( \omega_r (i - j) \big). \tag{46}$$

This confirms that the proposed factorization exactly recovers the target spectral component of the log-prior.

## D. Theoretical Motivation: Spectral Representation

Our choice of a spectral parameterization for the relative prior is grounded in Bochner's Theorem (Rasmussen & Williams, 2006), which provides the fundamental link between shift-invariant kernels and the frequency domain.

**Theorem 1** (Bochner). *A continuous, translation-invariant kernel $k : \mathbb{R}^d \times \mathbb{R}^d \to \mathbb{C}$ where $k(x, y) = \kappa(x - y)$ is positive definite if and only if $\kappa$ is the Fourier transform of a finite, non-negative Borel measure $\mu$. That is:*

$$\kappa(\delta) = \int_{\mathbb{R}^d} e^{-i\omega^\top \delta} d\mu(\omega). \tag{47}$$

In standard kernel methods (such as Gaussian Processes or Support Vector Machines) and linearized attention approximations (such as Performer), the kernel function represents a similarity metric and must strictly be positive definite. This constraint

forces the underlying spectral measure $\mu$ to be non-negative. Consequently, the learned weights in a Fourier approximation of such kernels must be positive, restricting the model to learning attractive interactions where similarity generally decays with distance.

GOAT circumvents this limitation by operating on the *log*-prior within the EOT objective. The attention distribution is derived as:

$$p_{ij} \propto \exp(s_{ij} + \mathcal{K}_{ij}). \tag{48}$$

Since the exponential function maps the real line to positive reals ($\mathbb{R} \to \mathbb{R}_{>0}$), the resulting prior $\pi_{ij} \propto \exp(\mathcal{K}_{ij})$ remains a valid probability distribution regardless of the sign of $\mathcal{K}_{ij}$.

This formulation parameterizes $\mathcal{K}_{ij}$ directly in log-space, avoiding the positivity constraints of Mercer kernels. Consequently, the spectral weights $\alpha_r$ and $\beta_r$ are free to take negative values. A positive weight corresponds to attraction (a peak at zero displacement), while a negative weight induces repulsion (a trough at zero displacement). This enables the model to actively suppress attention based on relative position, a capability mathematically inaccessible to standard positive-definite kernel approximations required by linear attention.

## E. Proofs of Stability Theorems

### E.1. Proof of Theorem 5.1 (Collapse to Prior)

*Proof.* The attention probability $p_{ij}$ is invariant to additive shifts in the score vector $s_i$. Let $s_{\min} = \min_k s_{ik}$ and define the shifted scores as $s'_{ik} = s_{ik} - s_{\min}$. By definition, for all $k$, we have:

$$0 \leq s'_{ik} \leq \omega_i. \tag{49}$$

The posterior probability is given by:

$$p_{ij} = \frac{\pi_{ij} \exp(s'_{ij})}{Z_i}, \quad \text{where } Z_i = \sum_k \pi_{ik} \exp(s'_{ik}). \tag{50}$$

We first derive global bounds for the partition function $Z_i$. Since $x \mapsto \exp(x)$ is monotonic, we can bound every term in the summation using the range of $s'_{ik}$:

$$Z_i \geq \sum_k \pi_{ik} \exp(0) = \sum_k \pi_{ik} = 1, \tag{51}$$

$$Z_i \leq \sum_k \pi_{ik} \exp(\omega_i) \tag{52}$$

$$= \exp(\omega_i) \sum_k \pi_{ik} = \exp(\omega_i). \tag{53}$$

We now bound the numerator, $N_{ij} = \pi_{ij} \exp(s'_{ij})$. Using the same range bounds:

$$\pi_{ij} \leq N_{ij} \leq \pi_{ij} \exp(\omega_i). \tag{54}$$

To find the lower bound for $p_{ij}$, we minimize the numerator and maximize the denominator:

$$p_{ij} = \frac{N_{ij}}{Z_i} \geq \frac{\pi_{ij}}{\exp(\omega_i)} = \pi_{ij} \exp(-\omega_i). \tag{55}$$

To find the upper bound for $p_{ij}$, we maximize the numerator and minimize the denominator:

$$p_{ij} = \frac{N_{ij}}{Z_i} \leq \frac{\pi_{ij} \exp(\omega_i)}{1} = \pi_{ij} \exp(\omega_i). \tag{56}$$

As $\omega_i \to 0$, both $\exp(\omega_i) \to 1$ and $\exp(-\omega_i) \to 1$. By the Squeeze Theorem, $\lim_{\omega_i \to 0} p_{ij} = \pi_{ij}$. $\square$

**E.2. Proof of Theorem 5.4 (Sensitivity Bounds)**

*Proof.* The sensitivity is $\Psi(\mathcal{C}) = 1 - p_{ij^\star}$. We analyze the probability $p_{ij^\star}$ in the limit $\omega_i \to 0$, where $s_{ij} \approx s_{ik}$ for all $j, k$, effectively vanishing from the softmax differences.

**Case 1 (Uniform):** The prior is constant, so $\mathcal{K}_{ij} = -\log L$. In the limit $\omega_i \to 0$, the total logits $z_{ij}$ become uniform. Thus $p_{ij^\star} = 1/L$.

$$\Psi_{\text{uni}} = 1 - \frac{1}{L} = \frac{L-1}{L} \xrightarrow{L \to \infty} 1. \tag{57}$$

**Case 2 (Peaked):** The prior ensures a margin in the log-space: $\mathcal{K}_{ij^\star} \geq \mathcal{K}_{ik} + \delta$ for all $k \in \mathcal{C}$. In the limit $\omega_i \to 0$, the content contributions vanish, and the total logit margin satisfies $z_{ij^\star} - z_{ik} \geq \delta$. The sink probability is bounded by:

$$p_{ij^\star} = \frac{1}{1 + \sum_{k \in \mathcal{C}} \exp(z_{ik} - z_{ij^\star})} \tag{58}$$

$$\geq \frac{1}{1 + \sum_{k \in \mathcal{C}} \exp(-\delta)} \tag{59}$$

$$= \frac{1}{1 + (L-1)\exp(-\delta)}. \tag{60}$$

The sensitivity is the complement:

$$\Psi_{\text{sink}} \leq 1 - \frac{1}{1 + (L-1)\exp(-\delta)} \tag{61}$$

$$= \frac{(L-1)\exp(-\delta)}{1 + (L-1)\exp(-\delta)} = \frac{L-1}{\exp(\delta) + L - 1}. \tag{62}$$

$\square$

# F. Priors vs. Positional Encodings: Isomorphism and Disentanglement

Throughout this work, we frame GOAT as learning a prior distribution, yet we benchmark it as a replacement for positional encodings and implement it via query key dot products. This raises a fundamental question regarding the theoretical status of the mechanism. Is an attention prior distinct from a positional encoding, or are they mathematically identical?

This section clarifies that while the two concepts are algebraically isomorphic in the final softmax operation, they represent fundamentally different parameterizations of the attention mechanism. This distinction between how the term is computed versus how it is derived is crucial for understanding the stability benefits of GOAT.

**F.1. Algebraic Isomorphism**

Let $z_{ij}$ denote the total logit entering the softmax function for query $i$ and key $j$.

From the perspective of positional encodings, one modifies the input vectors $\boldsymbol{q}$ and $\boldsymbol{k}$ via functions $\phi$ and $\psi$ such that the dot product encodes geometry. The logit is defined as follows:

$$z_{ij} = \langle \phi(\boldsymbol{q}, i), \psi(\boldsymbol{k}, j) \rangle \tag{63}$$

From the perspective of EOT, our derivation yields a logit composed of a raw content score plus a log prior term:

$$z_{ij} = \langle \boldsymbol{q}_c, \boldsymbol{k}_c \rangle + \mathcal{K}_{ij} \tag{64}$$

Because we implement $\mathcal{K}_{ij}$ by augmenting the query and key vectors with positional coordinates, as detailed in the parameterization section, GOAT is operationally a positional encoding. We have simply defined a specific transformation that reserves a subspace for position and ensures it interacts additively.

Therefore, we do not claim that priors and encodings are mutually exclusive implementations. Rather, we argue that the EOT formulation provides the derivation for a specific class of encodings that are additive, disentangled, and interpretable.

### F.2. Structural Entanglement

The utility of the prior framing becomes evident when analyzing how position interacts with content strength.

Standard encodings such as RoPE inject position multiplicatively via rotation. The total logit becomes:

$$z_{ij}^{\text{RoPE}} = (\boldsymbol{R}_i \boldsymbol{q})^\top (\boldsymbol{R}_j \boldsymbol{k}) \tag{65}$$

Notice that the magnitude of the positional signal is coupled to the magnitude of the semantic vectors. The positional contribution scales with $\|\boldsymbol{q}\|\|\boldsymbol{k}\|$. This creates structural entanglement. To express a strong positional preference, such as attending to the previous token, the model must increase the norm of the content vectors or align the content vectors specifically to maximize the dot product. This implies that the ability of the model to route information based on position is constrained by the semantic intensity of the tokens.

In contrast, the Prior derivation imposes an additive structure:

$$z_{ij}^{\text{GOAT}} = s_{ij}(\text{content}) + \mathcal{K}_{ij}(\text{structure}) \tag{66}$$

Here, the positional bias $\mathcal{K}_{ij}$ is independent of the content norms. The model can learn a strong structural preference that persists even when the semantic signal $s_{ij}$ is weak or zero. This decoupling explains the stability of GOAT in low signal regimes.

### F.3. Generalizing the Mechanism

We do not view attention priors as a theory of positional encodings that explains existing methods like RoPE. Rather, we view standard positional encodings as heuristic workarounds required when the attention mechanism is constrained to have a uniform prior.

If one forces $\mathcal{K}_{ij}$ to be constant, the only way to introduce geometry is to manipulate the cost function $s_{ij}$ by modulating the vectors $\boldsymbol{q}$ and $\boldsymbol{k}$. This leads to methods like RoPE. By relaxing this constraint and allowing $\mathcal{K}_{ij}$ to be learned, we obviate the need to manipulate the cost function.

Thus, GOAT is a generalization of the attention mechanism. It demonstrates that if the regularizer is sufficiently expressive, the encoding of position into content vectors becomes redundant. We benchmark against standard encodings to demonstrate this sufficiency, showing that a learnable prior is empirically superior to an entangled encoding.

## G. Why This Prior Parameterization Is Optimal Under Our Constraints

The EOT interpretation in Section 2 identifies attention as the solution of a KL-regularized transport problem, and therefore implies the existence of an additive log-prior term in the logits. It does not, by itself, determine how that log-prior should be parameterized. In this section we give a justification for the specific functional class used by GOAT. The conclusion is not that one can derive a universally optimal prior for all tasks, which would require specifying a data-generating distribution and a learning objective. Instead, we prove that once one imposes the constraints that are forced by our stated goals, namely kernel compatibility, translation equivariance of the relative component, and stability under length extrapolation, the admissible class of priors collapses to a finite-dimensional trigonometric family. We also prove that the ALiBi-style recency term is the unique maximum-entropy recency prior under a single moment constraint, and that key-only sink terms are the unique minimal-rank mechanism for query-independent defaults.

### G.1. Structural and Computational Constraints

Throughout we work on the integer line. Positions are indexed by integers, and we write $i$ for a query position and $j$ for a key position. We write $\mathcal{K}_{ij}$ for the additive log-prior in the attention logits, so that $z_{ij} = s_{ij} + \mathcal{K}_{ij}$ and $p_{i.} = \text{softmax}(z_{i.})$.

Our first constraint is imposed by the claim that GOAT can be realized in a single unmodified SDPA call without materializing an $L \times L$ bias matrix.

**Definition G.1** (SDPA-compatibility). A log-prior $\mathcal{K} : \mathbb{Z} \times \mathbb{Z} \to \mathbb{R}$ is *SDPA-compatible* with positional dimension $d_p < \infty$ if there exist functions

$$\varphi : \mathbb{Z} \to \mathbb{R}^{d_p}, \qquad \psi : \mathbb{Z} \to \mathbb{R}^{d_p} \tag{67}$$

such that for all $i, j \in \mathbb{Z}$,

$$\mathcal{K}_{ij} = \langle \varphi(i), \psi(j) \rangle. \tag{68}$$

The second constraint formalizes the intended semantics of a relative positional prior. The relative component should depend on displacement and not on absolute position.

**Definition G.2** (Translation equivariance of the relative log-prior). A log-prior $\mathcal{K}^{\mathrm{rel}}$ is *translation equivariant* if

$$\mathcal{K}^{\mathrm{rel}}_{i+a, j+a} = \mathcal{K}^{\mathrm{rel}}_{ij} \quad \text{for all } i, j, a \in \mathbb{Z}. \tag{69}$$

Equivalently, there exists a function $\kappa : \mathbb{Z} \to \mathbb{R}$ such that

$$\mathcal{K}^{\mathrm{rel}}_{ij} = \kappa(i - j) \quad \text{for all } i, j \in \mathbb{Z}. \tag{70}$$

The third constraint applies to the *learned* pattern-matching component. We require this component to be bounded to prevent it from arbitrarily dominating content logits as the context expands, while reserving unbounded trends (like recency slopes) for the fixed inductive bias term.

**Assumption G.3** (Bounded learned relative log-prior). *The learned displacement kernel $\kappa$ is bounded on $\mathbb{Z}$, that is,*

$$\sup_{d \in \mathbb{Z}} |\kappa(d)| < \infty. \tag{71}$$

We now show that these three constraints essentially determine the functional form of the relative prior.

### G.2. Classification of SDPA-Compatible Translation-Equivariant Relative Priors

The main result of this subsection is a classification theorem: bounded translation-equivariant priors that are realizable through a fixed finite-dimensional SDPA augmentation are necessarily finite trigonometric polynomials in the displacement. This statement is exact on $\mathbb{Z}$ and does not rely on periodic boundary conditions.

**Theorem 2** (Finite-dimensional SDPA compatibility forces a finite trigonometric relative prior). *Assume $\mathcal{K}^{\mathrm{rel}}$ is translation equivariant, so that $\mathcal{K}^{\mathrm{rel}}_{ij} = \kappa(i - j)$ for some $\kappa : \mathbb{Z} \to \mathbb{R}$. Assume further that $\mathcal{K}^{\mathrm{rel}}$ is SDPA-compatible with some finite positional dimension $d_p$, and that $\kappa$ is bounded as in Theorem G.3. Then there exist an integer $M \leq d_p$, angles $\theta_1, \ldots, \theta_M \in [0, 2\pi)$, and complex coefficients $c_1, \ldots, c_M \in \mathbb{C}$ such that for all $d \in \mathbb{Z}$,*

$$\kappa(d) = \sum_{m=1}^{M} c_m e^{i\theta_m d}. \tag{72}$$

*Moreover, since $\kappa$ is real-valued, one can group conjugate terms and obtain a purely real trigonometric representation: there exist $R \leq \lfloor d_p/2 \rfloor$, frequencies $\omega_1, \ldots, \omega_R \in (0, \pi]$, and real coefficients $\alpha_r, \beta_r \in \mathbb{R}$, along with a real constant $\gamma$, such that for all $d \in \mathbb{Z}$,*

$$\kappa(d) = \gamma + \sum_{r=1}^{R} \big( \alpha_r \cos(\omega_r d) + \beta_r \sin(\omega_r d) \big). \tag{73}$$

*Proof.* Let $d_p < \infty$ and let $\varphi, \psi : \mathbb{Z} \to \mathbb{R}^{d_p}$ be such that

$$\kappa(i - j) = \langle \varphi(i), \psi(j) \rangle \quad \text{for all } i, j \in \mathbb{Z}. \tag{74}$$

For each $j \in \mathbb{Z}$ define a sequence $f_j : \mathbb{Z} \to \mathbb{R}$ by $f_j(i) = \kappa(i - j)$. Then (74) implies

$$f_j(i) = \sum_{\ell=1}^{d_p} \varphi_\ell(i) \, \psi_\ell(j). \tag{75}$$

Hence every $f_j$ lies in the finite-dimensional subspace

$$U \triangleq \mathrm{span}\{\varphi_1, \ldots, \varphi_{d_p}\} \subset \mathbb{R}^{\mathbb{Z}}, \tag{76}$$

so the subspace

$$V \triangleq \operatorname{span}\{f_j : j \in \mathbb{Z}\} \tag{77}$$

satisfies $\dim(V) \le d_p$. Note that $\kappa(\cdot) = f_0(\cdot) \in V$.

Define the shift operator $T$ on sequences by $(Tg)(i) \triangleq g(i-1)$. Then for every $j \in \mathbb{Z}$ and every $i \in \mathbb{Z}$,

$$(Tf_j)(i) = f_j(i-1) = \kappa((i-1)-j) = \kappa(i-(j+1)) = f_{j+1}(i), \tag{78}$$

so $T(V) \subseteq V$. Since $T$ is invertible on $\mathbb{R}^{\mathbb{Z}}$ with inverse $(T^{-1}g)(i) = g(i+1)$, it follows that $T|_V$ is an invertible linear operator on the finite-dimensional space $V$.

We now work over $\mathbb{C}$ and consider the complexification $V_{\mathbb{C}} \triangleq V \otimes_{\mathbb{R}} \mathbb{C}$, on which $T$ acts $\mathbb{C}$-linearly. Because $V_{\mathbb{C}}$ is finite-dimensional, $T$ has a Jordan decomposition. Therefore there exist eigenvalues $\lambda_1, \ldots, \lambda_M \in \mathbb{C} \setminus \{0\}$ (not necessarily distinct) and a direct-sum decomposition of $V_{\mathbb{C}}$ into generalized eigenspaces. Write $\kappa \in V_{\mathbb{C}}$ as a sum of generalized eigencomponents,

$$\kappa = \sum_{m=1}^{M} v_m, \tag{79}$$

where each $v_m \in V_{\mathbb{C}}$ satisfies $(T - \lambda_m I)^{k_m} v_m = 0$ for some integer $k_m \ge 1$.

We claim that boundedness of $\kappa$ on $\mathbb{Z}$ forces every generalized eigencomponent to be a genuine eigenvector, and forces every eigenvalue to lie on the unit circle. To make this precise, fix a component $v$ satisfying $(T - \lambda I)^k v = 0$ with $\lambda \ne 0$. Define the forward difference operator $\Delta_\lambda \triangleq T - \lambda I$. The identity $(\Delta_\lambda)^k v = 0$ implies that the sequence $w(i) \triangleq \lambda^i v(i)$ has vanishing $k$th forward difference, hence is a polynomial in $i$ of degree at most $k-1$. We now prove this statement.

Define the standard forward difference $\Delta$ by

$$(\Delta u)(i) \triangleq u(i) - u(i-1). \tag{80}$$

Then

$$(\Delta w)(i) = w(i) - w(i-1) = \lambda^i v(i) - \lambda^{i-1} v(i-1) = \lambda^{i-1}\big(\lambda v(i) - v(i-1)\big) = -\lambda^{i-1}(\Delta_\lambda v)(i). \tag{81}$$

Iterating this identity shows that for every integer $r \ge 1$,

$$(\Delta^r w)(i) = (-1)^r \lambda^{i-r}\big((\Delta_\lambda)^r v\big)(i). \tag{82}$$

In particular, if $(\Delta_\lambda)^k v = 0$, then $(\Delta^k w) = 0$ on $\mathbb{Z}$. We now use the following lemma.

**Lemma G.4** (Sequences with vanishing $k$th forward difference are polynomials). *Let $w : \mathbb{Z} \to \mathbb{C}$ satisfy $\Delta^k w \equiv 0$ for some $k \ge 1$. Then there exists a polynomial $P$ of degree at most $k-1$ such that $w(i) = P(i)$ for all $i \in \mathbb{Z}$.*

*Proof.* For $t \ge 0$ define the binomial coefficient polynomial $\binom{i}{t}$ by

$$\binom{i}{t} = \frac{i(i-1)\cdots(i-t+1)}{t!}, \tag{83}$$

with the convention $\binom{i}{0} = 1$. These are polynomials in $i$ of degree $t$. They satisfy the exact identity

$$\Delta\binom{i}{t} = \binom{i-1}{t-1} \quad \text{for all } t \ge 1. \tag{84}$$

Iterating (84) yields $\Delta^t \binom{i}{t} = 1$ and $\Delta^{t+1}\binom{i}{t} = 0$.

We show that

$$\left\{ \binom{i}{0}, \binom{i}{1}, \ldots, \binom{i}{k-1} \right\} \tag{85}$$

forms a basis for the space of polynomial sequences of degree at most $k - 1$, and then show that any $w$ with $\Delta^k w = 0$ lies in their span. Consider the map $\Phi$ from the vector space spanned by these binomial polynomials into $\mathbb{C}^k$ defined by

$$\Phi(u) = (u(0), (\Delta u)(0), \dots, (\Delta^{k-1} u)(0)). \tag{86}$$

By the identities above, $\Phi(\binom{i}{t})$ has the form of a standard basis vector, specifically $(0, \dots, 0, 1, 0, \dots, 0)$ with the $1$ in the $(t+1)$st coordinate. Hence $\Phi$ is an isomorphism onto $\mathbb{C}^k$. Now let $w$ satisfy $\Delta^k w = 0$. Then the $k$ values $(w(0), (\Delta w)(0), \dots, (\Delta^{k-1} w)(0))$ determine $w$ uniquely by discrete integration, because for each $n$ one has

$$w(n) = w(0) + \sum_{m=1}^{n} (\Delta w)(m), \tag{87}$$

and similarly $(\Delta w)$ is determined by $(\Delta w)(0)$ and $(\Delta^2 w)$, and so on, terminating at $\Delta^{k-1} w$, which is constant since $\Delta^k w = 0$. Therefore there exists a unique $u$ in the span of $\{\binom{i}{t}\}_{t=0}^{k-1}$ with $\Phi(u) = \Phi(w)$, and uniqueness implies $u = w$ on all integers. Hence $w$ equals a polynomial of degree at most $k - 1$. $\qquad\square$

Applying Theorem G.4 to $w(i) = \lambda^i v(i)$ yields a polynomial $P$ of degree at most $k - 1$ such that

$$v(i) = \lambda^{-i} P(i) \quad \text{for all } i \in \mathbb{Z}. \tag{88}$$

Now invoke boundedness. If $|\lambda| > 1$, then $|\lambda^{-i}|$ grows without bound as $i \to -\infty$, and therefore $v$ is unbounded unless $P \equiv 0$. If $|\lambda| < 1$, then $|\lambda^{-i}|$ grows without bound as $i \to +\infty$, and again $v$ is unbounded unless $P \equiv 0$. Since $\kappa$ is bounded on $\mathbb{Z}$ and $\kappa$ is a finite sum of such components, every nonzero component must satisfy $|\lambda| = 1$.

Assume $|\lambda| = 1$ and $P$ has degree at least one. Then $|P(i)| \to \infty$ as $|i| \to \infty$, while $|\lambda^{-i}| = 1$ for all $i$, so $|v(i)| = |P(i)|$ is unbounded, contradicting boundedness. Therefore for every nonzero component, $P$ must be constant, which means $k = 1$ and $v$ is an eigenvector of $T$. Consequently each nonzero term in (79) has the form $v_m(i) = c_m \lambda_m^{-i}$ with $|\lambda_m| = 1$. Writing $\lambda_m = e^{-i\theta_m}$ yields $v_m(i) = c_m e^{i\theta_m i}$. Summing gives (72).

Finally, since $\kappa$ is real-valued, for each $\theta$ the coefficient of $e^{i\theta d}$ must be paired with the conjugate coefficient of $e^{-i\theta d}$ to produce a real sum. Grouping conjugate pairs yields (73) with real coefficients $\alpha_r, \beta_r$ and a possible constant term $\gamma$ corresponding to $\theta = 0$. This completes the proof. $\qquad\square$

**Consequences for parameterization.** Theorem 2 does not merely motivate a Fourier-like form. It states that under SDPA-compatibility, translation equivariance, and boundedness, the relative prior must be a finite trigonometric polynomial. In particular, using $R$ frequencies corresponds to choosing $d_p = 2R$ real dimensions, and (73) is the most general admissible form in that class.

We also need an exact dot-product realization of each trigonometric term.

**Proposition G.5** (Exact bilinear realization of a full Fourier mode). *Fix $\omega \in \mathbb{R}$ and $\alpha, \beta \in \mathbb{R}$. Define*

$$q_{\omega,\alpha,\beta}(i) = \begin{bmatrix} \alpha \cos(\omega i) + \beta \sin(\omega i) \\ \alpha \sin(\omega i) - \beta \cos(\omega i) \end{bmatrix}, \qquad k_\omega(j) = \begin{bmatrix} \cos(\omega j) \\ \sin(\omega j) \end{bmatrix}. \tag{89}$$

*Then for all $i, j \in \mathbb{Z}$,*

$$\langle q_{\omega,\alpha,\beta}(i), \, k_\omega(j) \rangle = \alpha \cos(\omega(i - j)) + \beta \sin(\omega(i - j)). \tag{90}$$

*Proof.* Expand the dot product and apply the angle-difference identities:

$$\cos(\omega(i - j)) = \cos(\omega i) \cos(\omega j) + \sin(\omega i) \sin(\omega j), \tag{91}$$

$$\sin(\omega(i - j)) = \sin(\omega i) \cos(\omega j) - \cos(\omega i) \sin(\omega j). \tag{92}$$

The claimed identity follows by direct substitution. $\qquad\square$

### G.3. Recency Priors and the ALiBi Term as a Maximum-Entropy Principle

In causal attention, for a fixed query position $i$, the admissible keys are $j \leq i$, hence the relevant structural variable is the lag $d = i - j \in \{0, 1, \ldots, i\}$. A recency prior is a distribution over lags. We now show that the exponential (geometric) family is uniquely singled out by a maximum-entropy principle (Jaynes, 1957) with a single moment constraint, and that its log-prior is equivalent to an ALiBi-style linear bias.

Fix a maximum context length $L \geq 1$ and consider lags $d \in \{0, 1, \ldots, L-1\}$. Let $\Delta_L$ denote the simplex

$$\Delta_L = \left\{ q \in \mathbb{R}^L : q_d \geq 0, \sum_{d=0}^{L-1} q_d = 1 \right\}. \tag{93}$$

Fix a target mean lag $\mu \in (0, L-1)$. Consider the constrained set

$$\mathcal{Q}_\mu = \left\{ q \in \Delta_L : \sum_{d=0}^{L-1} d\, q_d = \mu \right\}. \tag{94}$$

Define the Shannon entropy $H(q) = -\sum_{d=0}^{L-1} q_d \log q_d$ with the convention $0 \log 0 = 0$.

**Theorem 3** (Unique maximum-entropy recency prior under a mean constraint). *For each $\mu \in (0, L-1)$, the optimization problem*

$$\max_{q \in \mathcal{Q}_\mu} H(q) \tag{95}$$

*has a unique maximizer. This maximizer has the exponential form*

$$q_d^\star = \frac{e^{-\lambda d}}{\sum_{t=0}^{L-1} e^{-\lambda t}} \quad \text{for } d = 0, 1, \ldots, L-1, \tag{96}$$

*where $\lambda \in \mathbb{R}$ is the unique value such that $\sum_{d=0}^{L-1} d\, q_d^\star = \mu$.*

*Proof.* First note that $\mathcal{Q}_\mu$ is a nonempty compact convex subset of $\mathbb{R}^L$ because it is the intersection of the compact simplex $\Delta_L$ with an affine hyperplane. The entropy function $H$ is continuous on $\Delta_L$ and strictly concave on the relative interior of $\Delta_L$. Therefore $H$ attains its maximum on $\mathcal{Q}_\mu$, and any maximizer is unique because strict concavity implies that if $q^{(1)} \neq q^{(2)}$ are both maximizers, then

$$H\left(\tfrac{1}{2}(q^{(1)} + q^{(2)})\right) > \tfrac{1}{2}(H(q^{(1)}) + H(q^{(2)})), \tag{97}$$

contradicting maximality.

We now identify the maximizer by the method of Lagrange multipliers and verify that it lies in the interior. Let $q^\star$ be the unique maximizer. Suppose for contradiction that $q_{d_0}^\star = 0$ for some $d_0$. Because $\mu \in (0, L-1)$, the constraint $\sum d q_d = \mu$ forces positive mass on at least one lag $d_1 \neq d_0$. Consider a perturbation that transfers an infinitesimal amount of mass from $d_1$ to $d_0$ while preserving both constraints. One can choose such a perturbation within $\mathcal{Q}_\mu$ because the constraints are linear and $q^\star$ lies on a face of the simplex with at least two active coordinates. The directional derivative of $H$ in such a direction is $+\infty$ at a zero coordinate since $\partial_x(-x \log x) = -\log x - 1$ diverges as $x \to 0$, which contradicts optimality. Therefore $q_d^\star > 0$ for all $d$.

Since $q^\star$ lies in the interior, the KKT conditions reduce to stationarity of the Lagrangian. Introduce multipliers $\alpha, \beta \in \mathbb{R}$ for the normalization and mean constraints and consider

$$\mathcal{L}(q, \alpha, \beta) = -\sum_{d=0}^{L-1} q_d \log q_d + \alpha \left( \sum_{d=0}^{L-1} q_d - 1 \right) + \beta \left( \sum_{d=0}^{L-1} d\, q_d - \mu \right). \tag{98}$$

Stationarity with respect to $q_d$ gives, for each $d$,

$$\frac{\partial \mathcal{L}}{\partial q_d} = -(\log q_d + 1) + \alpha + \beta d = 0, \tag{99}$$

hence

$$\log q_d = \alpha - 1 + \beta d \quad \implies \quad q_d = e^{\alpha-1} e^{\beta d}. \tag{100}$$

Let $C = e^{\alpha-1} > 0$ and set $\lambda = -\beta$. Then $q_d = Ce^{-\lambda d}$. Enforcing $\sum_{d=0}^{L-1} q_d = 1$ yields

$$C^{-1} = \sum_{t=0}^{L-1} e^{-\lambda t}, \tag{101}$$

so $q^\star$ has the form (96). It remains to show existence and uniqueness of $\lambda$ satisfying the mean constraint. Define

$$Z(\lambda) = \sum_{t=0}^{L-1} e^{-\lambda t}, \qquad m(\lambda) = \sum_{d=0}^{L-1} d\, \frac{e^{-\lambda d}}{Z(\lambda)}. \tag{102}$$

Then $m(\lambda)$ is continuous and strictly decreasing in $\lambda$. To see strict monotonicity, note that $m(\lambda)$ is the expectation of $d$ under an exponential family with natural parameter $-\lambda$, and one has

$$m'(\lambda) = -\mathrm{Var}_{q^\star(\lambda)}(d) < 0, \tag{103}$$

since the variance is positive for any distribution with full support on $\{0, \dots, L-1\}$ and $L \geq 2$. Moreover, $\lim_{\lambda \to +\infty} m(\lambda) = 0$ and $\lim_{\lambda \to -\infty} m(\lambda) = L-1$. Therefore for every $\mu \in (0, L-1)$ there exists a unique $\lambda$ with $m(\lambda) = \mu$. $\qquad\square$

The relevance to attention is that (96) implies that the optimal log-prior over lags is affine in the lag:

$$\log q_d^\star = -\lambda d - \log Z(\lambda). \tag{104}$$

Under causal masking, $d = i - j \geq 0$, so for fixed $i$,

$$\log q_{i-j}^\star = -\lambda(i-j) - \log Z(\lambda) = \lambda j + c_i, \tag{105}$$

where $c_i = -\lambda i - \log Z(\lambda)$ is constant across admissible keys $j \leq i$.

**Proposition G.6** (Equivalence of lag-linear and key-linear biases under causal masking). *Fix a query index $i$ and consider keys $j \leq i$. Let $m \in \mathbb{R}$. Define two bias functions*

$$\mathcal{B}_{ij}^{\mathrm{lag}} \triangleq -m(i-j), \qquad \mathcal{B}_{ij}^{\mathrm{key}} \triangleq mj. \tag{106}$$

*Then $\mathcal{B}_{ij}^{\mathrm{lag}} = \mathcal{B}_{ij}^{\mathrm{key}} - mi$ for all $j \leq i$. Consequently, for any content logits $s_{ij}$,*

$$\mathrm{softmax}_{j \leq i}\big(s_{ij} + \mathcal{B}_{ij}^{\mathrm{lag}}\big) = \mathrm{softmax}_{j \leq i}\big(s_{ij} + \mathcal{B}_{ij}^{\mathrm{key}}\big). \tag{107}$$

*Proof.* The identity

$$\mathcal{B}_{ij}^{\mathrm{lag}} = -m(i-j) = mj - mi \tag{108}$$

is immediate. Softmax is invariant to adding a constant to all logits in a fixed row: for any vector $x$ and scalar $c$,

$$\mathrm{softmax}(x + c\mathbf{1}) = \mathrm{softmax}(x). \tag{109}$$

Apply this with $c = -mi$ to the row indexed by $i$. $\qquad\square$

Theorem 3 and Theorem G.6 provide a justification for the ALiBi-style term used in our implementation. Among all recency priors on lags with a fixed mean, the least-committal prior is exponential in the lag, and under causal masking this is equivalent to a key-linear bias.

### G.4. Key-Only Defaults and Attention Sinks as Minimal Rank

The EOT perspective in Section 5 shows that when content evidence is weak, the posterior collapses toward the prior. For stability in that regime it is useful to allow the prior to place substantial mass on a small set of default keys. We now show that the most direct way to express a query-independent default preference is a key-only log-prior $u(j)$, and that this mechanism is minimal in a precise rank sense.

Fix a finite length $L$ and identify indices with $\{0, 1, \ldots, L - 1\}$. Let $u \in \mathbb{R}^L$ and define the matrix $U \in \mathbb{R}^{L \times L}$ by

$$U_{ij} \triangleq u_j. \tag{110}$$

**Theorem 4** (Key-only priors are exactly rank-one and require one positional lane). *The matrix $U$ satisfies* $\operatorname{rank}(U) \leq 1$. *If $u \neq 0$, then* $\operatorname{rank}(U) = 1$. *Moreover, $U$ admits an SDPA-compatible representation with $d_p = 1$ by taking $\varphi(i) \equiv 1$ and $\psi(j) = u_j$. Conversely, if a matrix $M \in \mathbb{R}^{L \times L}$ has identical rows, meaning that $M_{ij}$ depends only on $j$, then $M$ is of the form $U$ for some $u$ and has rank at most one.*

*Proof.* Let $\mathbf{1} \in \mathbb{R}^L$ be the all-ones vector. Then $U = \mathbf{1}u^\top$, which is an outer product, hence $\operatorname{rank}(U) \leq 1$. If $u \neq 0$, then $\mathbf{1}u^\top$ is a nonzero outer product, hence has rank exactly one. The SDPA-compatible representation with $d_p = 1$ is immediate from $U_{ij} = 1 \cdot u_j = \langle \varphi(i), \psi(j) \rangle$ with $\varphi(i) \equiv 1$ and $\psi(j) = u_j$. For the converse, if $M_{ij} = m(j)$ for some function $m$, then $M = \mathbf{1}m^\top$, hence rank at most one, and it equals $U$ with $u = m$. $\square$

### G.5. Synthesis: Optimality Within the Admissible Class

We now summarize what has been proved. The EOT derivation fixes the form of the posterior as a softmax of content logits plus an additive log-prior. The requirement that this prior be implemented inside a single SDPA call forces a bilinear form in positional features, as in Theorem G.1. If we further require that the relative component be translation equivariant and stable under extrapolation, then Theorem 2 implies that the only possible bounded relative priors in this SDPA-compatible class are finite trigonometric polynomials in the displacement. Theorem G.5 shows that our query rotation construction is an exact bilinear realization of the most general such Fourier mode, including the antisymmetric component that encodes directionality.

For causal language models, a recency bias can be justified independently by a maximum-entropy principle. Theorem 3 shows that among all recency priors with a fixed mean lag, the unique entropy maximizer is exponential in the lag. Theorem G.6 then shows that, under causal masking, this yields an ALiBi-equivalent key-linear bias. Finally, Theorem 4 shows that query-independent defaults, including attention sinks, correspond exactly to rank-one key-only log-priors and require exactly one additional positional lane. This is the minimal SDPA-compatible mechanism for controlling sink behavior without entangling content representations.

These results justify our parameterization as optimal in the following sense: under the constraints that are forced by our goals, namely SDPA compatibility, translation equivariance of the relative component, and boundedness for stability, the admissible family of relative priors is exactly the finite trigonometric class, and our construction realizes the general element of that class with minimal per-frequency dimension. Under an information-theoretic principle for recency in causal attention, the unique least-committal recency prior is exponential in lag, which is ALiBi-equivalent, and our key-linear slope term is therefore a principled component of the log-prior. Under the requirement of query-independent defaults, the key-only sink term is the unique minimal-rank mechanism.

# H. C4 Language Modeling Setup

*Table 2.* **C4-125M Training Setup.**

| Hyperparameter | Value |
|---|---|
| Dataset | C4 (English), `allenai/c4` (streaming), `train`/`validation` splits |
| Tokenizer | `gpt2` (HF fast) |
| Context Length | $L_{\text{train}} = 2048$ |
| GOAT Configuration | $d_{\text{pos}} = 12$ (Pos Rank 2, Abs Rank 8), Key Bias enabled |
| Training Budget | $4.0 \times 10^9$ tokens |
| Architecture | GPT-style decoder-only Transformer (pre-norm) |
| Model Size | 12 layers, 12 heads, $d_{\text{model}} = 768$ ($\approx$125M parameters) |
| Optimizer | AdamW ($\beta_1 = 0.9, \beta_2 = 0.95, \epsilon = 10^{-8}$) |
| Learning Rate | Peak $1 \times 10^{-4}$, cosine decay to $3 \times 10^{-5}$ |
| Warmup | 2000 steps |
| Weight Decay | 0.1 |
| Batch Size | 32 |
| Precision | `bf16` mixed precision |
| Gradient Clipping | 1.0 |

# I. Ablations and Convergence Details

*Table 3.* **Fourier Mode Ablation on C4.** Increasing the number of modes mainly improves extrapolation, while in-distribution perplexity stays stable.

| $R$ | $d_p$ | PPL 2k | PPL 4k | PPL 8k |
|---|---|---|---|---|
| 1 | 4 | 35.49 | 34.81 | 38.28 |
| 2 | 6 | 35.52 | 34.95 | 38.64 |
| 4 | 10 | 35.49 | 34.57 | 35.97 |
| 8 | 18 | **35.18** | **34.15** | **35.15** |

*Table 4.* **Frequency Grid Ablations on C4.** The learned coefficients compensate for broad changes in the fixed geometric grid, so GOAT is insensitive to choice of frequency.

| Frequency Setting | PPL 2k | PPL 4k | PPL 8k |
|---|---|---|---|
| Base 1,000 | 35.48 | 34.88 | 38.42 |
| Base 10,000 | 35.52 | 34.95 | 38.64 |
| Base 100,000 | 35.62 | 35.04 | 38.51 |
| Learned frequencies | **35.39** | **34.74** | **37.99** |

*Table 5.* **Initialization Ablation on C4.** Zero initialization is reliable, random initialization is slightly worse, and an ALiBi warm start gives a small extrapolation gain.

| Initialization | PPL 2k | PPL 4k | PPL 8k |
|---|---|---|---|
| Zero | 35.52 | 34.95 | 38.64 |
| Random | 35.54 | 35.11 | 38.88 |
| ALiBi warm start | 35.56 | **34.93** | **38.48** |

*Table 6.* **Learned Spectral Structure in the 1.23B FineWeb Model.** Positional structure is concentrated in a subset of heads and is mostly directional.

| Statistic | Value |
| --- | --- |
| Frequency energy split | 13% / 45% / 31% / 11% across frequency indices from lowest to highest |
| Per-head amplitude | Min 0.14, max 3.99, mean 1.09, std 0.80 |
| Active heads | 287 of 704 heads have total spectral amplitude above 1.0 |
| Directional heads | 662 of 704 heads have larger antisymmetric than symmetric coefficients; median $|\beta|/|\alpha| = 1.84$ |
| Layer trend | Layer 0: mean amplitude 2.04, 28/32 active heads; layer 21: mean amplitude 0.70, 5/32 active heads |

*Table 7.* **Training Loss Checkpoints.** GOAT does not introduce slower early optimization at either the 125M C4 scale or the 1.23B FineWeb scale.

| Run | Step | Tokens | RoPE Loss | GOAT Loss |
| --- | --- | --- | --- | --- |
| C4 125M | 500 | 131M | 6.777 | **6.623** |
| C4 125M | 2,000 | 524M | 4.829 | **4.648** |
| C4 125M | 5,000 | 1.31B | 4.090 | **4.052** |
| C4 125M | 15,000 | 3.93B | 3.535 | **3.520** |
| FineWeb 1.23B | 500 | 131M | 5.627 | **5.537** |
| FineWeb 1.23B | 2,000 | 524M | 3.952 | **3.871** |
| FineWeb 1.23B | 5,000 | 1.31B | 3.370 | **3.351** |
| FineWeb 1.23B | 10,000 | 2.62B | 3.003 | **2.993** |

## J. FineWeb Language Modeling Setup

*Table 8.* **FineWeb Training Setup.**

| Hyperparameter | Value |
| --- | --- |
| Dataset | FineWeb, streaming |
| Tokenizer | GPT-2 BPE |
| Context Length | $L_{\text{train}} = 2048$ |
| GOAT Configuration | Pos Rank 4, Abs Rank 8, Key Bias enabled, base 10k |
| Training Budget | $\approx 3.13 \times 10^9$ tokens at the selected GOAT checkpoint |
| Architecture | Llama-style decoder-only Transformer |
| Model Size | 22 layers, 32 heads, $d_{\text{model}} = 2048$, $d_{\text{ff}} = 5632$ ($\approx$1.23B parameters) |
| Optimizer | AdamW ($\beta_1 = 0.9, \beta_2 = 0.95, \epsilon = 10^{-8}$) |
| Learning Rate | Peak $3 \times 10^{-4}$, cosine decay to $3 \times 10^{-5}$ |
| Warmup | 2000 steps |
| Weight Decay | 0.1 |
| Batch Size | 128 sequences (262k tokens/step) |
| Precision | `bf16` mixed precision, TF32 enabled |
| Gradient Clipping | 1.0 |

## K. Spatial VQA Training Configuration

*Table 9.* **Spatial VQA Results.** Test scenes use image-token scan orders not seen during training; GOAT's mixed 2D image and 1D text prior improves both answer NLL and exact answer accuracy.

| Method | Answer NLL ↓ | Answer Acc. ↑ |
| --- | --- | --- |
| No PE | 0.359 | 17.2% |
| RoPE | 0.367 | 16.5% |
| GOAT | **0.350** | **20.3%** |

*Table 10.* **Spatial VQA Training Setup.** The benchmark tests whether a model can answer spatial questions when test scenes use image-token scan orders not seen during training.

| Hyperparameter | Value |
|---|---|
| Dataset | CIFAR-10 scenes with four images arranged on a canonical $2\times2$ grid |
| Task | Answer-only spatial VQA with absolute-position and relation questions |
| Scan Orders | Scene-order families with held-out test orders |
| Image Tokens | $4\times4$ patches, k-means codebook with 512 visual tokens |
| Train / Test Scenes | 50k / 10k |
| Sequence Length | 324 prediction positions |
| Architecture | GPT-style decoder-only Transformer |
| Model Size | 12 layers, 8 heads, $d_{\text{model}} = 416$, $d_{\text{ff}} = 1664$ |
| GOAT Configuration | Pos Rank 8, Abs Rank 8, Key Bias enabled, base 10k, image grid $16\times16$ |
| Optimizer | AdamW ($\beta_1 = 0.9, \beta_2 = 0.95, \epsilon = 10^{-8}$) |
| Learning Rate | Peak $3 \times 10^{-4}$, cosine decay to $3 \times 10^{-5}$ |
| Warmup | 500 steps |
| Weight Decay | 0.1 |
| Batch Size | 64 sequences |
| Precision | `bf16` mixed precision, TF32 enabled |
| Model Selection | Best checkpoint by validation answer NLL |

## L. ImageNet Training Configuration

*Table 11.* **ImageNet-1k Training Setup.**

| Hyperparameter | Value |
|---|---|
| Dataset | ImageNet-1k (ILSVRC 2012) |
| Architecture | ViT-Small (SwiGLU variant) |
| Model Size | 12 layers, 6 heads, $d_{\text{model}} = 384$, patch $16\times16$ |
| GOAT Configuration | $d_{\text{pos}} = 32$ (16 pairs), 2D factorization, Key Bias enabled |
| Input Resolution | $224 \times 224$ (training) |
| Optimizer | AdamW ($\beta_1 = 0.9, \beta_2 = 0.999$) |
| Learning Rate | Peak $1 \times 10^{-3}$, cosine decay |
| Weight Decay | 0.05 |
| Training Schedule | 150 epochs |
| Warmup | 20 epochs |
| Batch Size | 1024 |
| Augmentation | RandAugment ($N = 2, M = 9$) |
| Mixup / CutMix | Mixup ($\alpha = 0.8$), CutMix ($\alpha = 1.0$) |
| Regularization | Label Smoothing ($\epsilon = 0.1$), Random Erasing ($p = 0.25$) |
| Stochastic Depth | Drop Path rate 0.1 (linear decay) |
| Precision | `float16` mixed precision |

