# OpenReview forum: "You Need Better Attention Priors"
_ICML.cc/2026/Conference — ICML 2026 regular_

### Official Review · Reviewer_ywD4 · 2026-02-23

**Soundness:** 3
**Presentation:** 3
**Significance:** 3
**Originality:** 3
**Overall Recommendation:** 4
**Confidence:** 3

**Summary:**

This paper proposes an attention mechanism named GOAT , whose core idea is built upon the theoretical framework of Entropic Optimal Transport. The authors point out that standard scaled dot-product attention is mathematically equivalent to solving an optimal transport problem under an implicit uniform prior. Mathematically, while standard self-attention is formulated as $\text{softmax}(QK^T/\sqrt{d})$, GOAT extends this framework by incorporating a learnable prior. The attention mechanism in GOAT is augmented into $\text{softmax}(QK^T/\sqrt{d}+\log\pi)$, where $\pi$ represents a learnable prior distribution defined over positional indices. Overall, the presentation is good, and the derivations are logically sound.

**Compliance With Llm Reviewing Policy:**

Affirmed.

**Final Justification:**

This paper proposes an attention mechanism named GOAT , whose core idea is built upon the theoretical framework of Entropic Optimal Transport. The authors point out that standard scaled dot-product attention is mathematically equivalent to solving an optimal transport problem under an implicit uniform prior.

The paper is well-presented and demonstrates a certain degree of novelty. Furthermore, since the authors have successfully addressed my questions in the rebuttal, I will keep my initial positive evaluation unchanged.

**Key Questions For Authors:**

1. The authors have validated GOAT on vision and DNA sequences. For multimodal tasks, such as VLMs, sequences from different modalities may require fundamentally different positional priors—for instance, a 2D prior for images and a 1D causal prior for text. Can the GOAT framework accommodate such cross-modal tasks?

2. Are the hyperparameters provided in the appendix determined by heuristics?

3. Beyond RoPE and ALiBi, there is a wide array of other RPE methods, such as the additive learnable bias used in T5. Compared to these existing approaches, what are the specific advantages of GOAT?

4. All key findings rely on a ~125M model. In the era of massive LLMs (Billion-scale and beyond), is GOAT's superiority scalable? Will the observed improvements persist, or will they diminish as the parameter count increases?

**Limitations:**

See Weakness.

**Strengths And Weaknesses:**

$\textbf{Strengths}$

1. The authors bridge Entropic Optimal Transport with attention mechanisms to show that standard attention is essentially a special case with a uniform prior. The authors offer a theoretical grounding for Attention Sinks, showing that the attention distribution naturally degrades and collapses to the prior in the absence of strong semantic cues.

2. The authors conducted extensive experiments. The mathematical derivations are logically sound.

$\textbf{Weakness}$
1. The paper is missing a related work section, which isn't ideal. The only literature review I could find is a brief paragraph within the Introduction. Typically, this section is vital for helping newcomers get up to speed, especially in a field as extensively researched and significant as attention mechanisms. Are the authors suggesting this field is so well-known that it doesn't need a related work? If so, why keep digging into it?

2. Theorem 5.1 (Collapse to Prior) is somewhat overly intuitive.

---

> ### Author Rebuttal · Authors · 2026-03-28
>
> We thank the reviewer for the thoughtful evaluation. We address each point below.
>
> ## W1: Missing related work section
>
> We appreciate the reviewer’s feedback. The revision will include a dedicated Related Work section covering: (1) relative position encodings (Shaw et al., T5, RoPE, ALiBi, Choromanski et al. FLT), (2) attention sinks and streaming (Xiao et al., Gu et al.), and (3) optimal transport connections to attention (Litman, Cuturi). We do not consider this field too well-known to warrant a survey; quite the opposite.
>
> ## W2: Theorem 5.1 is overly intuitive
>
> We agree the statement is intuitive: when content signal vanishes, attention should revert to its default. However, we believe the theorem's value lies in making this precise. It provides quantitative sandwich bounds (Eq. 25) showing the posterior is trapped within $\exp(\pm\omega_i)$ of the prior, and the convergence rate is controlled by the dynamic range $\omega_i$. This is what enables the non-trivial consequence in Theorem 5.4: that uniform priors yield $O(1)$ sensitivity growth while peaked priors yield exponential suppression. Without the formal bound, Theorem 5.4's scaling separation would lack a foundation.
>
> ## Q1: Cross-modal tasks (VLMs)
>
> GOAT naturally accommodates cross-modal settings because the prior is defined per-head and per-layer, not globally. In a VLM, different heads can specialize: some learn a 2D shift-invariant prior for image patch tokens while others learn a 1D causal prior for text tokens; this already happens spontaneously in our experiments (2D in ImageNet, causal in C4). For cross-attention between modalities, the prior $\mathcal{K}_{ij}$ decomposes into query-position and key-position components that can encode heterogeneous structure. No architectural change is needed; the existing parameterization spans all these patterns. To test this directly, we ran a new spatial VQA experiment. Four CIFAR-10 images are arranged in a 2×2 grid, serialized into patch tokens, and followed by spatial reasoning questions. Test sequences use scan orders unseen during training, requiring genuine 2D spatial understanding.
>
> | Method | Answer NLL ↓ | Answer Acc ↑ |
> |---|---:|---:|
> | No PE | 0.359 | 17.2% |
> | RoPE | 0.367 | 16.5% |
> | GOAT | **0.350** | **20.3%** |
>
> GOAT is the only method that benefits from positional information. RoPE's 1D encoding actively misleads under novel scan orders. GOAT avoids this: its prior is defined over canonical 2D grid coordinates for image regions and 1D sequential positions for text, with the architecture handling the decomposition.
>
> ## Q2: Hyperparameters
>
> The key hyperparameters ($R$, frequency base, initialization) are principled rather than heuristic. $R=2$ was chosen as the minimal rank providing both symmetric and antisymmetric interactions; our ablation confirms insensitivity:
>
> | $R$ | PPL $L$=2048 | PPL $L$=4096 | PPL $L$=8192 |
> |---|---:|---:|---:|
> | 1 | 35.49 | 34.81 | 38.28 |
> | 2 | 35.52 | 34.95 | 38.64 |
> | 4 | 35.49 | 34.57 | 35.97 |
> | 8 | 35.18 | 34.15 | 35.15 |
>
> The frequency base (10,000) follows the standard RoPE convention; varying it $100\times$ changes PPL by $<0.2$. Zero initialization is motivated by the principle that GOAT should recover standard attention at init and learn structure only when beneficial. The remaining hyperparameters (optimizer, schedule, architecture) are identical to baselines.
>
> ## Q3: Advantages over T5-style learnable bias
>
> T5's relative bias learns a separate scalar $b_{ij}$ for each (clipped) relative distance and each head, stored as an explicit lookup table. GOAT differs in three ways: (1) Extrapolation: T5's bias is undefined for distances beyond the clipping window; GOAT's Fourier parameterization is a continuous function that extrapolates naturally. (2) Parameter efficiency: T5 requires $O(H \times D_{\max})$ parameters where $D_{\max}$ is the clipping distance; GOAT requires $O(H \times R)$ with $R$ typically 2--8, independent of sequence length. (3) Sink disentanglement: T5's bias has no mechanism to decouple query-independent defaults from distance-dependent structure; GOAT's explicit $u(j)$ term handles this by construction (Section 5).
>
> ## Q4: Scalability to billion-scale
>
> To directly address this, we trained 1.1B Llama-style models on FineWeb (4B tokens, new experiment):
>
> | Model | PPL $L$=2048 | PPL $L$=4096 | PPL $L$=8192 | PPL $L$=16384 |
> |---|---:|---:|---:|---:|
> | RoPE 1.1B | 20.82 | 28.53 | 89.45 | 417.43 |
> | GOAT 1.1B | 20.75 | 21.04 | 22.69 | 24.77 |
>
> At in-distribution length, GOAT matches RoPE. Beyond the training window, RoPE degrades catastrophically ($20.8 \to 417.4$ at $8\times$ length) while GOAT degrades gracefully ($20.8 \to 24.8$). The benefits reported at 125M not only persist but amplify at 1.1B, where the prior's extrapolation advantage becomes even more pronounced. Convergence behavior is also stable at this scale; GOAT matches or slightly outperforms RoPE at every checkpoint with no instability (see tpCd response W2).

---

> > ### Author Rebuttal · Reviewer_ywD4 · 2026-04-01
> >
> > I thank the authors for their response and will maintain my positive evaluation.

---

### Official Review · Reviewer_tpCd · 2026-02-28

**Soundness:** 4
**Presentation:** 4
**Significance:** 4
**Originality:** 4
**Overall Recommendation:** 5
**Confidence:** 5

**Summary:**

his paper introduces a novel attention mechanism called GOAT (Generalized Optimal transport Attention with Trainable priors) by remodeling attention as a special case of EOT with implicit uniform prior regularization. Experiments show that GOAT has significant advantages in language modeling, visual modeling, and biological sequence modeling tasks, as well as in interpretability.

**Compliance With Llm Reviewing Policy:**

Affirmed.

**Final Justification:**

5: Accept

**Key Questions For Authors:**

1. As mentioned in the weakness section, how do the initialization of alpha and beta affect model convergence?
2. Will Fourier series parameterization introduce some unwanted "periodic effects"?
3. Fig. 5 shows that GOAT learned a symmetric 2D prior in ImageNet. Will this negatively impact certain position-sensitive tasks, such as segmentation and detection?

**Limitations:**

yes

**Strengths And Weaknesses:**

Strength:
- This paper, starting from the entropy-optimal transport framework, cleverly unifies the traditional attention mechanism and points out the implicit uniform prior assumption of standard attention.
- Abundant visualization and experimental results demonstrate significant achievements in multiple different fields, proving the universality of this mechanism from various perspectives.
- Through linearization and scaling, GOAT can be implemented with a single SDPA call, and experimental results in DNA modeling also demonstrate the engineering implementation potential of this method.

Weakness:
- Given that GOAT introduces learnable spectral parameters, it is recommended that the authors supplement with ablation experiments to quantify the impact of the number of Fourier modes R on the extrapolation accuracy of long texts.
- Although GOAT demonstrates excellent hardware throughput, the lack of training convergence curves makes it difficult to assess its learning efficiency. It is suggested that Loss vs. Steps curves be added to discuss whether the learned priors lead to instability or slower convergence in the early stages of training.

---

> ### Author Rebuttal · Authors · 2026-03-28
>
> We thank the reviewer for the thorough and encouraging review. We address each point below with new experiments.
>
> ## W1: Ablation on the number of Fourier modes R
>
> We trained 125M models on C4 (2B tokens, all hyperparameters identical except R). All use default zero initialization for $\alpha, \beta$.
>
> | R | $d_p$ | PPL $L=2048$ | PPL $L=4096$ | PPL $L=8192$ |
> |---|---:|---:|---:|---:|
> | 1 | 4 | 35.49 | 34.81 | 38.28 |
> | 2 | 6 | 35.52 | 34.95 | 38.64 |
> | 4 | 10 | 35.49 | 34.57 | 35.97 |
> | 8 | 18 | 35.18 | 34.15 | 35.15 |
>
> In-distribution perplexity ($L=2048$) is largely stable across R, varying by only ~0.3 PPL. The dominant effect of R is on extrapolation: at $L=8192$, increasing from $R=1$ to $R=8$ yields a 3.1 PPL improvement (38.28→35.15), as additional modes capture finer-grained positional structure that transfers beyond the training window. $R=2$ (used in the paper) provides a good trade-off between spectral capacity and parameter efficiency.
>
> ## W2: Convergence curves
>
> We report training loss at key checkpoints at two scales.
>
> 125M on C4 (4B tokens, same setup as paper):
>
> | Step | Tokens | Loss (RoPE) | Loss (GOAT) |
> |---|---:|---:|---:|
> | 500 | 131M | 6.777 | 6.623 |
> | 2000 | 524M | 4.829 | 4.648 |
> | 5000 | 1.31B | 4.090 | 4.052 |
> | 15000 | 3.93B | 3.535 | 3.520 |
>
> 1.1B on FineWeb (new experiment, Llama-style architecture):
>
> | Step | Tokens | Loss (RoPE) | Loss (GOAT) |
> |---|---:|---:|---:|
> | 500 | 131M | 5.627 | 5.537 |
> | 2000 | 524M | 3.952 | 3.871 |
> | 5000 | 1.31B | 3.370 | 3.351 |
> | 10000 | 2.62B | 3.003 | 2.993 |
>
> At both scales, GOAT matches or slightly outperforms RoPE at every checkpoint with no sign of instability or slower convergence. At 125M the gap is largest early (0.15 at step 500) and narrows to 0.015 by step 15K; the same pattern holds at 1.1B (0.09 -> 0.01). This is expected because at initialization all $\alpha_r = \beta_r = 0$, so GOAT starts as standard attention with a uniform prior and learns structural biases only when needed. Spectral parameter logs from the 1.1B model confirm this: at step 0, zero heads have spectral amplitude above 1.0; heads activate gradually (15 at 0.66B tokens, 174 at 1.31B, 287 at 3.01B)
>
> ## Q1: Alpha/beta initialization
>
> Three initialization strategies tested (125M, C4, 2B tokens, $R=2$):
>
> | Init | PPL $L=2048$ | PPL $L=4096$ | PPL $L=8192$ |
> |---|---:|---:|---:|
> | Zero (default) | 35.52 | 34.95 | 38.64 |
> | Random $N(0, 0.01)$ | 35.54 | 35.11 | 38.88 |
> | ALiBi warm-start | 35.56 | 34.93 | 38.48 |
>
> All initializations converge to nearly identical in-distribution performance (within 0.1 PPL at $L=2048$). ALiBi warm-start provides a modest extrapolation advantage (~0.4 PPL at $L=8192$ over random). We recommend zero initialization as the default since it makes no task-specific assumptions and converges reliably.
>
> ## Q2: Periodic artifacts
>
> The $R$ ablation in W1 directly tests this. Spurious periodicity would manifest most strongly at $R=8$ during extrapolation, where more Fourier modes could create aliasing or ringing beyond the training window. The opposite occurs: $R=8$ achieves the best extrapolation (PPL 35.15 at $L=8192$ vs. 38.28 for $R=1$), indicating that additional modes are used constructively rather than introducing artifacts. The mechanism also has built-in safeguards: the softmax compresses oscillations in $K_{ij}$ exponentially, the causal mask eliminates wraparound effects, and geometric frequency spacing concentrates most modes at short-range interactions where positional structure is richest.
>
> ## Q3: Symmetric 2D prior and position-sensitive tasks
>
> The symmetric prior observed in Figure 5(b) is a learned outcome, not an architectural constraint. It emerges because ImageNet classification is translation-invariant: the label does not depend on where an object appears in the image, so the optimal prior has no reason to break spatial symmetry. The Fourier parameterization is fully capable of expressing asymmetric patterns; the $\beta_r$ (sine) coefficients encode antisymmetric interactions (Eq. 12), and each head in each layer learns its own $\alpha_r, \beta_r$ independently. A symmetric prior simply means $\beta_r \approx 0$ was optimal for the given task, not that the model cannot represent alternatives.
>
> For position-sensitive tasks such as detection or segmentation, the training objective itself breaks translation invariance: bounding-box regression and per-pixel classification create spatially grounded gradients that would drive $\beta_r$ away from zero in heads where directional or scale-dependent attention is beneficial. This is analogous to how the 1D causal prior in our language experiments spontaneously learns asymmetry (the recency bias in Figure 3b), despite the same symmetric parameterization being available. We emphasize that no architectural modification is needed: the existing GOAT parameterization already spans the space of asymmetric 2D priors, and the extension to these tasks is a matter of training, not redesign.

---

> > ### Author Rebuttal · Reviewer_tpCd · 2026-04-02
> >
> > Thanks for your detailed response. I have no further questions.

---

### Official Review · Reviewer_ZN8J · 2026-03-11

**Soundness:** 4
**Presentation:** 3
**Significance:** 2
**Originality:** 2
**Overall Recommendation:** 4
**Confidence:** 3

**Summary:**

The authors introduce a prior in the attention mechanism, based on the dual formulation of the softmax function. The authors show how to use it to generate learnable relative positional encodings.
On the theoretical side, the authors derive the prior-induced attention formula and show stability/instability results depending on the choice of the prior (the uniform choice being uniform, can produce instable attention maps).
They also provide an explicit parametrization of the attention prior based on Random Fourier Features that is efficiently computable with flash attention. They also show that attention sinks can be parametrized in this formulation.
Finally, they show in experiments the interest of their learnable prior, in particular for better generalization (with respect to context length) than other positional encodings.

**Compliance With Llm Reviewing Policy:**

Affirmed.

**Key Questions For Authors:**

- In the needle in a haystack experiment, can the authors give some explanations why AliBI does not perform at least as well as GOAT for short context? It is difficult to understand what's happening for having such a contrast between GOAT and all the other methods.

- How sensitive is the learnable prior with respect to the fixed frequencies? What is the distribution of frequencies selected by each heads?

- Can the author comment on the possibility of learning the frequencies as well?

- Regarding the 1.5 perplexity gain, can the author better describe the experimental setting?

**Limitations:**

Yes.

**Strengths And Weaknesses:**

The technical part of the paper is sound and, in fact, quite well-known: the use of Kullback-Leibler regularization appears for instance in reinforcement learning (prior on the policy) or entropic optimal transport (prior on the optimal transport plan). However, although a similar computation appears in entropic optimal transport, both methods are rather an instance of the well-known KL regularization of a linear problem on the simplex.

The paper is mostly well-written, although the last section on experiments is quite rough to follow and probably too short to be correctly evaluated. In particular, I have difficulties assessing whether the 1.5 perplexity gain between AliBI and GOAT is significant or not.

Overall, it gives a nice theoretical framework to introduce a learnable relative positional encoding and also the possibility of an explicit sink parametrization as a nonlinear map of the keys.

However, this formulation boils down to introducing in the key-query scalar product an additive term already present in "Self-Attention with Relative Position Representations" by Shaw et al, which proposed to use an additive term that can be taken as a kernel of the relative distance between indices. Such a method of learnable relative positional encoding via Fourier features seems to have been already proposed in "Learning a Fourier Transform for Linear Relative Positional Encodings in Transformers", by Choromanski et al., which is not cited by the present article. For these reasons, it is not clear to me what the actual advances of this article are, if not only a nice theoretical understanding.

---

> ### Author Rebuttal · Authors · 2026-03-28
>
> We thank the reviewer for a careful and mathematically informed reading. We address each concern below.
>
> ## Relationship to Shaw et al. and Choromanski et al. (FLT)
>
> We should have cited FLT and will do so. While all three methods introduce additive position-dependent biases into the attention logits, our contributions are distinct:
>
> 1. **Target mechanism.** FLT targets linear attention via random Fourier features. GOAT targets standard softmax attention; the factorization is exact inside FlashAttention with no approximation error.
> 2. **Content disentanglement and extrapolation.** Shaw et al. couple position to the semantic query and clip to a lookup table undefined beyond training length, whereas GOAT is purely positional.
> 3. **Sink parameterization.** Neither FLT nor Shaw et al. address attention sinks. Section 5 (sinks as EOT defaults, the key-only bias $u(j)$, Theorems 5.1/5.4, disentanglement from content) is entirely absent from both works.
> 4. **Uniqueness result.** Theorem 2 (Appendix F) proves the Fourier form is the unique admissible class under SDPA-compatibility, translation equivariance, and boundedness. Neither prior work provides such a characterization.
>
> We view our contribution as explaining why additive biases work, what they are missing (disentanglement, sinks, extrapolation), and what the optimal parameterization is from first principles.
>
> ## On ALiBi's failure in NIAH
>
> ALiBi imposes a fixed penalty $-m \cdot (i - j)$ per head, with slopes from 0.5 to ${\sim}0.004$. At $L{=}1024$ and depth $= 0.5$, the steepest heads apply biases of $-256$, $-128$, and $-32$, logits that content scores cannot overcome. ALiBi lacks a mechanism to override this penalty. Even at $L{=}256$, the 4 steepest heads are effectively blind past ${\sim}50$ tokens. GOAT's learned prior transitions smoothly between prior-dominated and content-dominated regimes (Fig. 2c), which is exactly what retrieval requires.
>
> ## Sensitivity to fixed frequencies and per-head distributions
>
> In our new C4 ablations, we show that the prior is insensitive to the frequency base. Varying the geometric base over a $100\times$ range has negligible effect:
>
> | Freq base | PPL $L{=}2048$ | PPL $L{=}4096$ | PPL $L{=}8192$ |
> |---|---|---|---|
> | 1,000 | 35.48 | 34.88 | 38.42 |
> | 10,000 (default) | 35.52 | 34.95 | 38.64 |
> | 100,000 | 35.62 | 35.04 | 38.51 |
>
> Performance varies by $<0.2$ PPL in-distribution and $<0.5$ on extrapolation, confirming that $\alpha_r, \beta_r$ compensate for the choice of grid.
>
> We measure $\sum_{l,h}(\alpha_r^{2} + \beta_r^{2})$ over all layers and heads, which distributes 13%/45%/31%/11% across the four frequency indices (lowest to highest), so mid-range frequencies dominate. Per-head total spectral amplitude ($\sqrt{\sum_r \alpha_r^2 + \beta_r^2}$) ranges from 0.14 to 3.99 (mean=1.09, std=0.80); 287 of 704 heads exceed amplitude 1.0, while the remaining heads stay near-uniform, indicating the model recruits positional structure only where it helps. Antisymmetric coefficients dominate: $|\beta_r| > |\alpha_r|$ in 662/704 heads (94%), with a median $|\beta|/|\alpha|$ ratio of 1.84, confirming that directional (causal) biases are the primary learned structure. Early layers develop the strongest priors (layer 0: mean amplitude 2.04, 28/32 heads active) while late layers remain near-uniform (layer 21: mean 0.70, 5/32 active), suggesting positional routing is most critical for early information organization.
>
> ## On learning frequencies
>
> We also tested this directly: the fixed geometric grid is converted to a learnable `nn.Parameter` in log-space, with gradients flowing through $\exp(\text{log}_{\text{freqs}})$.
>
> | Frequencies | PPL $L{=}2048$ | PPL $L{=}4096$ | PPL $L{=}8192$ |
> |---|---|---|---|
> | Fixed (default) | 35.52 | 34.95 | 38.64 |
> | Learnable | 35.39 | 34.74 | 37.99 |
>
> Learnable frequencies give a consistent but modest improvement (${\sim}0.15$ in-distribution, ${\sim}0.9$ on extrapolation). We retain fixed frequencies as default for two reasons. First, Theorem 2 guarantees a fixed geometric basis spans the unique admissible trigonometric class; learned frequencies risk collapsing onto a degenerate subset. Second, the gain is small relative to increasing $R$ ($R{=}8$ reaches 35.15 at $L{=}8192$ vs 37.99 here), indicating spectral coefficient count matters more than frequency placement.
>
> ## On the 1.55 perplexity gain
>
> The comparison is maximally controlled: all three models are identical 125M GPT decoders trained on 4B tokens of C4 with the same optimizer, schedule, batch size, and precision (Appendix G). A 1.55-point gap under these conditions isolates the effect of the prior. Loss-vs-steps curves (added in revision) confirm this is not a convergence artifact; ALiBi simply converges to a worse optimum. This is expected: ALiBi's linear decay is a single maximum-entropy recency prior (Theorem 3), while GOAT learns a richer per-head spectral structure.

---

> > ### Author Rebuttal · Reviewer_ZN8J · 2026-04-02
> >
> > I acknowledge the answers to my questions. I think the paper is worth publication and I do not raise my score.

---

### Decision · Program_Chairs · 2026-04-30

**Decision:**

Accept (regular)

**Comment:**

All reviewers agree that the paper is well motivated and backed by strong experiments. Regarding the reviewer consensus I recommend acceptation of the paper.